# Tau binding protein CAPON induces tau aggregation and neurodegeneration

Shoko Hashimoto [1], Yukio Matsuba[1], Naoko Kamano[1], Naomi Mihira[1], Naruhiko Sahara[2], Jiro Takano[1], Shin-ichi Muramatsu [3,4], Takaomi C. Saido [1] & Takashi Saito [1,5]

To understand the molecular processes that link Aβ amyloidosis, tauopathy and neurodegeneration, we screened for tau-interacting proteins by immunoprecipitation/LC-MS. We identified the carboxy-terminal PDZ ligand of nNOS (CAPON) as a novel tau-binding protein. CAPON is an adaptor protein of neuronal nitric oxide synthase (nNOS), and activated by the N-methyl-D-aspartate receptor. We observed accumulation of CAPON in the hippocampal pyramidal cell layer in the $App^{NL-G-F}$-knock-in (KI) brain. To investigate the effect of CAPON accumulation on Alzheimer's disease (AD) pathogenesis, CAPON was overexpressed in the brain of $App^{NL-G-F}$ mice crossbred with *MAPT* (human tau)-KI mice. This produced significant hippocampal atrophy and caspase3-dependent neuronal cell death in the CAPON-expressing hippocampus, suggesting that CAPON accumulation increases neurodegeneration. CAPON expression also induced significantly higher levels of phosphorylated, oligomerized and insoluble tau. In contrast, CAPON deficiency ameliorated the AD-related pathological phenotypes in tauopathy model. These findings suggest that CAPON could be a druggable AD target.

[1] Laboratory for Proteolytic Neuroscience, RIKEN Center for Brain Science, 2-1 Hirosawa, Wako-City, Saitama 351-0198, Japan. [2] Department of Functional Brain Imaging Research, National Institute of Radiological Sciences, National Institutes for Quantum and Radiological Science and Technology, 4-9-1 Anagawa, Inage-ku, Chiba-City, Chiba 263-8555, Japan. [3] Division of Neurology, Jichi Medical University, 3311-1 Yakushiji, Shimotsuke-City, Tochigi 329-0498, Japan. [4] Center for Gene & Cell Therapy, The Institute of Medical Science, The University of Tokyo, Tokyo 108-8639, Japan. [5] Department of Neuroscience and Pathobiology, Research Institute of Environmental Medicine, Nagoya University, Nagoya-City, Aichi 464-8601, Japan. Correspondence and requests for materials should be addressed to S.H. (email: shoko.hashimoto@riken.jp) or to T.C.S. (email: saido@brain.riken.jp) or to T.S. (email: takashi.saito.aa@riken.jp)

Alzheimer's disease (AD) is the most common neurode-generative disorder and the major cause of dementia[1]. It is also a progressive disease, the symptoms of which worsen over decades. The neuropathological hallmarks of AD include extracellular deposits of amyloid-β (Aβ), the major component of senile plaques, and neurofibrillary tangles (NFTs) composed of hyperphosphorylated tau protein. According to the amyloid cascade hypothesis, the initial elevation of Aβ levels is the unique primary trigger of AD[2], with a deterioration in Aβ metabolism and the accumulation of Aβ plaques starting two decades prior to the appearance of clinical symptoms[3]. In contrast, tauopathy, which is observed in a variety of neurodegenerative disorders, is significantly enhanced following Aβ amyloidosis[3]. However, the molecular mechanisms that link Aβ amyloidosis, tauopathy and neurodegeneration remain unresolved[4].

We previously developed two lines of novel AD model mice based on an *App* (amyloid precursor protein) knock-in (KI) strategy[5]. The first mouse model ($App^{NL-F/NL-F}$ KI; $App^{NL-F}$) carries two mutations (KM670/671NL: Swedish; I716F: Iberian) identified in human familial AD. The second model ($App^{NL-G-F/NL-G-F}$; $App^{NL-G-F}$) carries an additional mutation (E639G: Arctic mutation), and exhibits aggressive pathology[5,6]. Both lines, which produce humanized Aβ without overexpressing APP, show pronounced Aβ amyloidosis, gliosis, and memory deficits[5,7]. Further, we established human *MAPT* (human tau)-KI mice (hTau-KI), which express 6 isoforms of wild-type (WT) human tau instead of murine tau. Although we thought that the double-KI mice generated by cross-breeding *App* KI and hTau-KI mice might show greater AD pathology than the single *App*-KI mice due to tau humanization, we noted no overt pathological changes in the former mice. These results suggested the presence of mechanisms/factors in addition to Aβ amyloidosis that induce tauopathy and neurodegeneration.

In order to understand the molecular mechanism of tau accumulation, we screened for tau-interacting proteins using Wtau-transgenic (Tg) mice, which express WT human tau tagged with a Flag epitope[8]. We isolated tau-binding proteins by immunoprecipitation using a Flag-tag antibody and identified them by LC-MS/MS analysis. The methods used to generate the tau interactome were validated by identification of the tubulin beta-4A chain as one of the tau-binding proteins (Supplementary Data 1), given that tau is a microtubule-binding protein.

Among the proteins identified in the tau interactome, we focused on one protein, named carboxy-terminal PDZ ligand of neuronal nitric oxide synthase (CAPON). CAPON is an adaptor protein of neuronal nitric oxide synthase (nNOS), which acts as an enzyme for the production of nitric oxide (NO), and is involved in N-methyl-D-aspartate (NMDA) receptor-mediated excitotoxicity[9]. CAPON is generally assumed to recruit substrates to nNOS, and facilitate their NO-mediated modification through protein-protein interactions[9]. Several studies have reported that CAPON polymorphisms are associated with schizophrenia, and other psychiatric disorders[10,11]. Moreover, Richier et al,[12] demonstrated that CAPON positively regulates spine density, while Li et al,[13] reported that CAPON regulates neuronal cell death downstream of the NMDA receptor. It therefore appears that CAPON contributes to neurotransmission and neuronal excitotoxicity. In addition, according to a report by Hashimoto et al[14], CAPON is upregulated in CA1 pyramidal cells in the AD brain. These results imply that CAPON may play an important role in the pathogenesis of AD, although the underlying mechanism(s) remain unknown.

To clarify the effect of CAPON on AD pathology, we used an adeno-associated virus (AAV)-mediated approach to introduce CAPON cDNA into the brain of $App^{NL-G-F}$ and $App^{NL-G-F}$/*MAPT* (hTau) double-KI mice which we newly developed. Given

that the hTau-KI mouse expresses an endogenous level of WT human tau, we were able to analyze the effects of the human tau protein. Our results revealed that CAPON expression facilitates hippocampal atrophy, with accompanying neuronal cell death. We also verified that deficiency of CAPON in P301S-Tau-Tg tauopathy mouse model suppressed tau pathology and neurodegeneration. In addition, we examined the molecular mechanisms that lead to CAPON-induced neuronal cell death and AD pathology, i.e. tau phosphorylation and aggregation, Aβ deposition, and gliosis, in CAPON-expressing mice.

## Results

**Generation and charactirization of human *MAPT* knockin mouse.** In this study, to evaluate functions of a novel tau-binding protein: CAPON on AD-related pathologies including tau pathology, we used a new mouse model expressing tau protein in the manner of human brain.

Normal adult human brains express six distinct isoforms which are classified into 3-repeats (3R)-tau and 4-repeats (4R)-tau depending on the number of repeated microtubule-binding domains. On the other hand, adult mouse possesses 4R-tau only. Importantly, NFTs in human AD comprise an equal mixture of all 3R and 4R tau isoforms. Therefore, mouse model should express all tau isoforms when we evaluate formation of tau pathology. In addition, ideally, the mouse model of tau pathology should be based on the KI strategy because overexpression of tau may disturb the normal physiological functions of neurons, such as microtubule assembly and synaptic functions. Accordingly, we generated human *MAPT* KI (hTau-KI) mice, in which the entire human *Mapt* gene was inserted at the murine *Mapt* gene locus (Supplementary Fig. 1). Wild-type mice predominantly expressed 4R tau, while *MAPT* KI mice expressed all human tau isoforms as observed in human samples (Fig. 1a and Supplementary Fig. 2a, b). The relative ratio of mRNA for 4R-tau/3R-tau was 0.69 ± 0.07 (Fig. 1b, c), which is close to that of human brain. *MAPT* KI mouse did not display accerolated neuroinflammation, neuronal cell death and brain atrophy (Supplementary Fig. 2c, d). Moreover, cross-breeding of *MAPT* KI with $App^{NL-G-F}$ KI did not alter amyloid pathology, neuroinflammation, and neuronal cell death of $App^{NL-G-F}$ KI (Fig. 1d, e). These findings indicate that humanization of the *Mapt* gene does not affect neurodegenerative processes.

**AD pathology alters the expression pattern of CAPON.** In order to identify tau-binding proteins, we generated a tau interactome, based on mass spectrometry-based immunoprecipitation proteomics. We performed immunoprecipitation with a Flag-tag antibody using brain lysate from WT (negative control) and Wtau-tg mice, which express WT human tau tagged with a Flag epitope, to isolate tau and its binding proteins. Supplementary Data 1 summarizes the proteins which were specifically identified from the Wtau-tg mice. We subsequently focused on CAPON as it is specifically expressed in the brain[9], and polymorphisms have been identified in several psychiatric diseases[10]. Moreover, CAPON is also upregulated in the hippocampal pyramidal cells of AD patients[14], and may therefore play a pivotal role in the etiology of this disease.

According to Hashimoto et al[14], although CAPON accumulates in hippocampal pyramidal cells in the AD brain, the overall amount of the protein is significantly lowered in AD patients compared with healthy controls. We therefore first examined whether Aβ amyloidosis and tau pathology affect the level or pattern of CAPON expression in the brain. Western blot analyses of hippocampal and cortical tissues from 12–14-month-old WT, $App^{NL-G-F}$ (with Aβ pathology), hTau single-KI (without Aβ pathology) and $App^{NL-G-F}$ / hTau double-KI (with Aβ pathology)

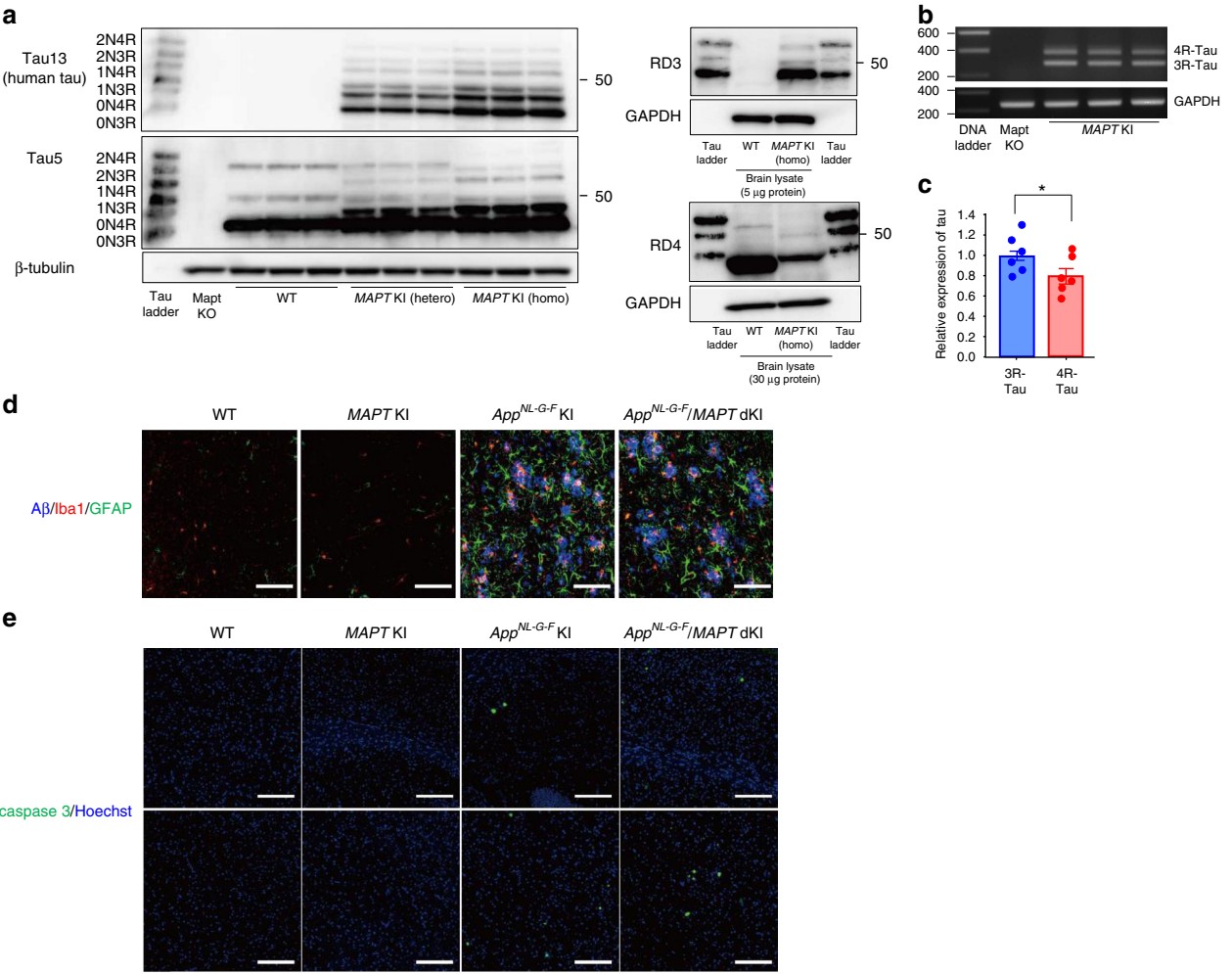

**Fig. 1** Characterization of human *MAPT* KI mice. **a** Tau isoforms in *MAPT* KI mouse brains were determined. Brain extracts derived from 3-month-old WT, heterozygous (hetero) and homozygous (homo) mice were subjected to western blot analysis after dephosphorylation (*n* = 3 each). **b**, **c** The relative expression of 3R and 4R tau were calculated by semi-quantitative RT-PCR. Data reperesent mean ± SEM (*n* = 6, Stutent *t* test (t(10) = 1.883, *p = 0.0446). **d** Amyloid pathology and neuroinflammation were detected in 24-month-old WT, single *MAPT* KI, *App*[NL-G-F] single-KI and *App*[NL-G-F] /*MAPT* double-KI mice. The brain sections were triple-immunostained using 82E1 (blue), anti-GFAP (green) and anti-Iba1 antibody (red). Scale bar: 100 μm. **e** The brain sections of 24-month-old WT, *MAPT* single-KI, *App*[NL-G-F] single-KI and *App*[NL-G-F] /*MAPT* double-KI mice were immunostained by cleaved-caspase 3 antibody. Scale bar: 100 μm. Source data are provided as a Source Data file

mice revealed that presence of with Aβ pathology resulted in higher CAPON expression levels in these animals (Fig. 2a and Supplementary Fig. 4a, b). Intriguingly, immunohistochemical analyses showed a significantly stronger CAPON signal in the pyramidal cell layer of *App*[NL-G-F] mice than in the WT mice (Fig. 2b). These results indicate that Aβ pathology elevates the level and localization of CAPON in hippocampal pyramidal cells. On the other hand, P301S-Tau-Tg mice (tauopathy model) exhibited an age-dependent decline in CAPON protein (Fig. 2c), suggesting that severe pathology such as that associated with frontotemporal dementia with parkinsonism-17 (FTDP-17) mutant tau-induced cytotoxicity decreases the level of CAPON, presumably because the number of neuronal cells where CAPON is mainly expressed is reduced (Supplementary Fig. 6). Indeed, P301S-Tau-Tg mice display severe brain atrophy in association with neuronal cell death, in an age-dependent manner[15].

How does Aβ pathology alter the expression pattern of CAPON? We selected two candidates as potential mediators of this change: nNOS and neuroinflammation. Further analysis revealed a dramatic reduction in the protein levels of CAPON, without any change in its mRNA levels, in the nNOS-deficient

mouse brain, suggesting that stabilization of the CAPON protein is highly dependent on its interaction with nNOS (Supplementary Fig. 5a, b, c). Because pathogenic Aβ species interact with and activate NMDA receptor[16,17], nNOS is considered to be activated under amyloid pathology[18]. Therefore, we quantified the interactions between CAPON and nNOS in the WT and *App*[NL-G-F] mouse brain using the Duolink system[19]. In this system, if nNOS is located in a neighboring distance from CAPON, optical signals will be detected as dots. As shown in Fig. 3a, b, we observed no difference in the levels of nNOS protein or in the CAPON-nNOS interaction between WT and *App*[NL-G-F] tissue, indicating that interaction with nNOS does not contribute to CAPON accumulation in the *App*[NL-G-F] mouse.

We next examined the involvement of neuroinflammation. Shao et al[20] reported that lipopolysaccharide (LPS) stimulation induces CAPON expression, and Cheng et al[21] proposed that CAPON increases after spinal cord injury. Therefore, neuroinflammation could also be involved in the elevation of CAPON expression under amyloid pathology. Indeed, *App*[NL-G-F] mice display marked activation of microglia and astrocytes[5]. Similar to Shao et al, we examined the expression pattern of CAPON 1 week

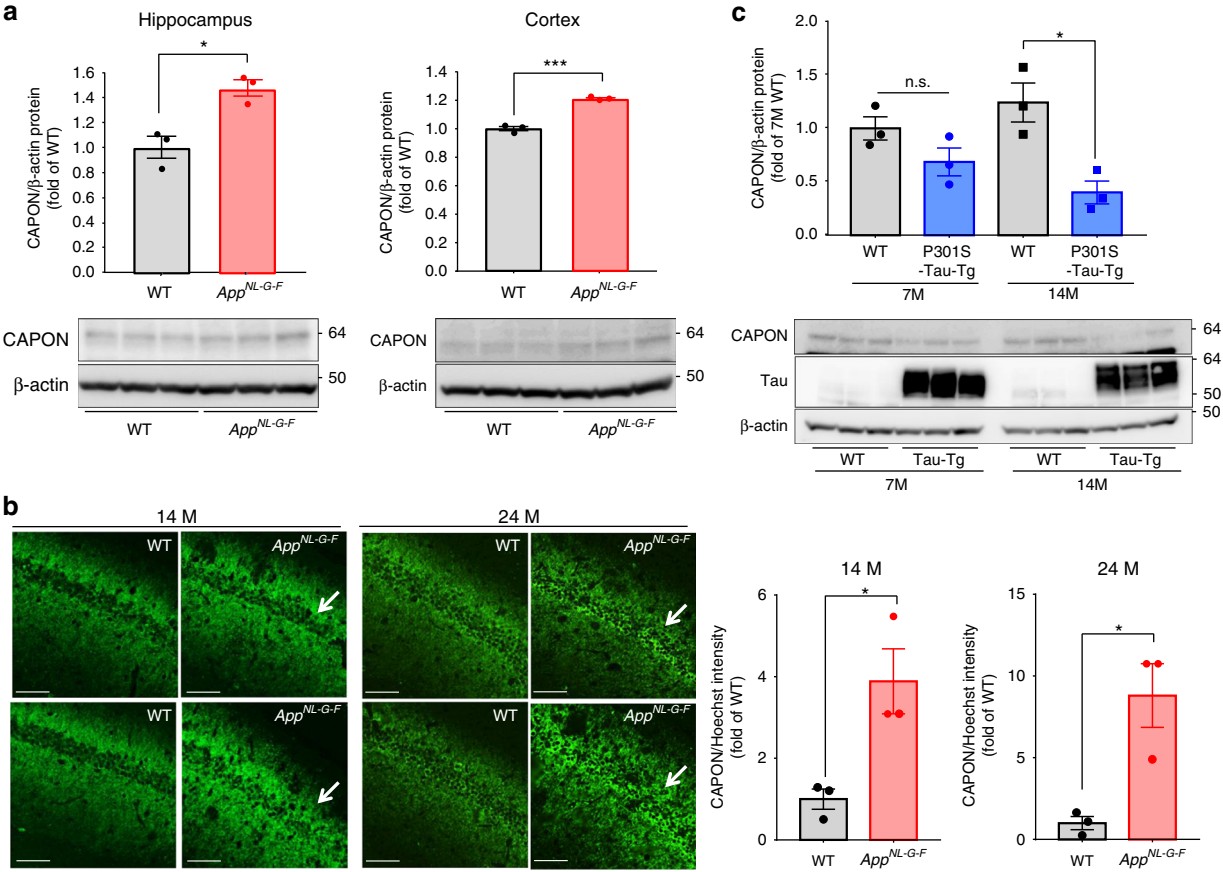

**Fig. 2** Endogenous CAPON expression in $App^{NL-G-F}$ mice. **a** The protein levels of CAPON in the hippocampi and cortices of 14-month-old (M) WT and male $App^{NL-G-F}$ mice were determined. Values shown in the graph represent the band intensity of CAPON divided by the intensity of β-actin, expressed as the mean level ± SEM ($n = 3$, *$p < 0.05$, ***$p < 0.001$). **b** Representative images of the hippocampal CA1 region of 14- and 24-month-old WT and male $App^{NL-G-F}$ mice immunohistochemically stained for CAPON. Arrows indicate the hippocampal pyramidal cell layer. The fluorescence intensity in the pyramidal cell layer is also quantitatively represented as the mean intensity level ± SEM ($n = 3$, *$p < 0.05$). Scale bar: 100 μm. **c** The protein level of CAPON in the hippocampi of male 7- and 14-month-old WT and P301S-Tau-Tg mice was determined. The values shown in the graph are the band intensity of CAPON divided by the intensity of β-actin, with the results expressed as the mean relative expression level of CAPON ± SEM ($n = 3$, *$p < 0.05$). Source data are provided as a Source Data file

after intraventricular injection of LPS. Immunohistochemical analyses revealed significant induction of CAPON in regions where the microglia was highly activated (Fig. 3c-left). However, the CAPON signals did not colocalize with Iba1 (Fig. 3c-left, and Supplementary Fig. 7) and non-neuronal cells markers (Supplementary Fig. 7), whereas the signals colocalized with the nNOS immunoreactivity (Fig. 3c-right), and other neuronal cell markers (Supplementary Fig. 7). Therefore, CAPON appears partially increased in neuronal rather than glial cells. These results indicate that neuroinflammation associated with Aβ pathology may possibly be involved in CAPON elevation.

**Expression of CAPON cDNA induces neuronal cell death-mediated hippocampal atrophy.** We then hypothesized that the increase in CAPON protein in $App^{NL-G-F}$ mice aggravates the pathological events that occur in response to Aβ pathology, i.e. tauopathy and neurodegeneration. To investigate whether CAPON inflicts damage on the brain, we evaluated the effect of CAPON cDNA expression on AD pathology in $App^{NL-G-F}$ / hTau double-KI mice. Humanization of tau itself did not change CAPON levels in hippocampus and cortex, but cross-breeding with $App^{NL-G-F}$ elevated CAPON levels (Supplementary Fig.4), suggesting that amyloid pathology increased CAPON level also in hTau-KI mice.

AAV was used to introduce the CAPON or GFP gene, expressed under the control of the synapsin I promoter (Supplementary Fig. 8a), bilaterally into the ventricles of 12-month-old double-KI mice (Supplementary Fig. 8b, c). This produced strong gene expression, particularly in the hippocampus (Supplementary Fig. 8d, e). To assess brain atrophy, magnetic resonance imaging (MRI) was performed 7 days and 3 months after AAV injection (Supplementary Fig. 8c). Whereas the AAV-GFP-injected mice did not show any significant difference in brain volume between the two time-points, the AAV-CAPON-injected mice exhibited a large decrease in hippocampal volume at 3 months, falling to approximately 77% of the control (Fig. 4a, b). The rate of reduction in the hippocampal volume correlated negatively with the expression level of CAPON (Supplementary Fig. 13a).

We also examined hippocampal atrophy and neuronal cell death using histochemical analyses. Significant shrinkage of the hippocampus was confirmed by hematoxylin and eosin (H&E) staining of brain sections (Fig. 4c). We also observed a significant decrease in NeuN (neuron marker) -positive neuronal cells and an increase in cleaved caspase 3 (cell death marker) signals in the hippocampus of CAPON-expressing mice (Fig. 4d). The signals of cleaved caspase 3 colocalized with neuronal cell markers (Fig. 4e). Moreover, TUNEL-positive cells were also detected in

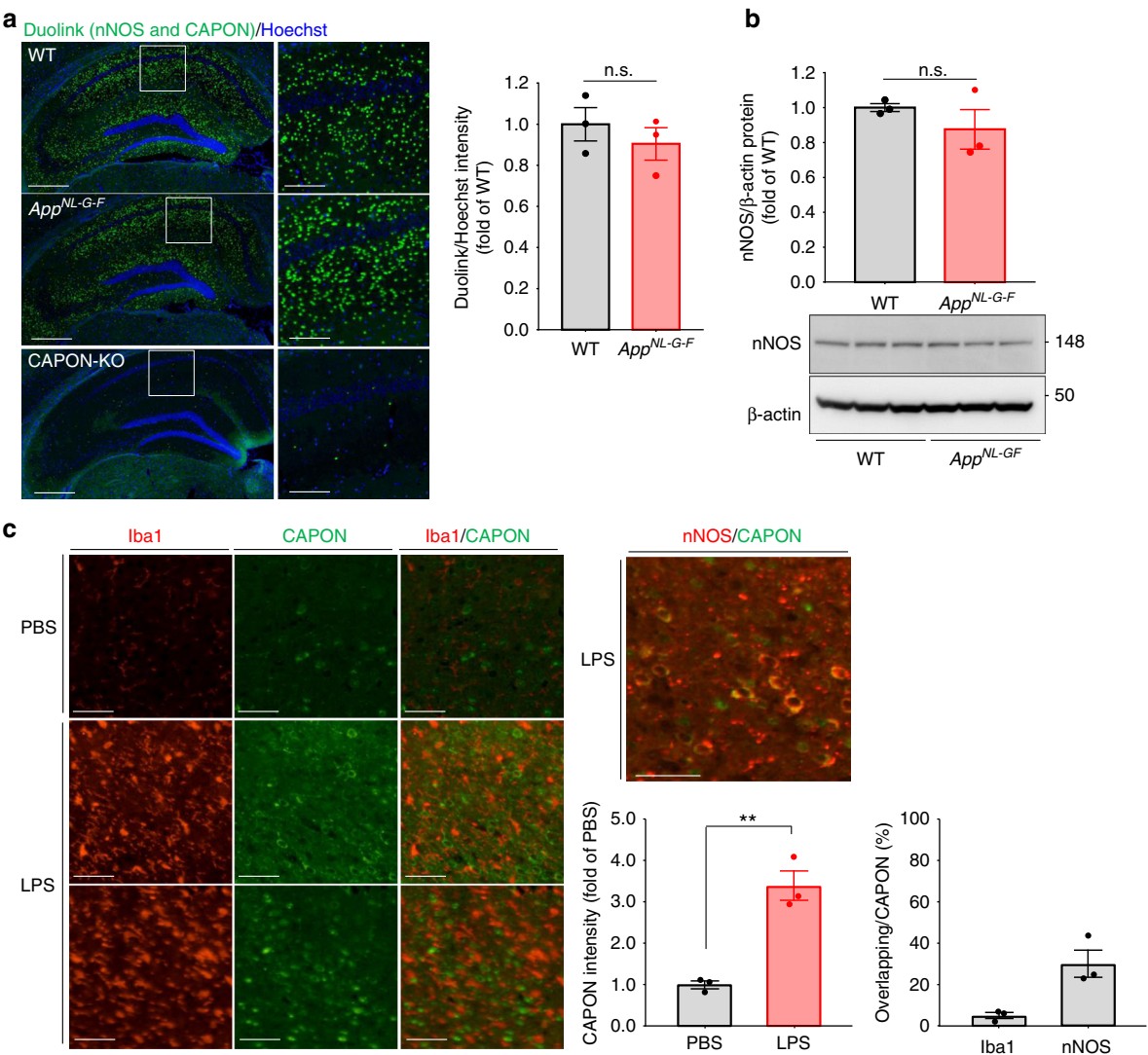

**Fig. 3** Neuroinflammation and interaction with nNOS could be involved in CAPON expression. **a** Duolink signals (green) for the nNOS-CAPON interaction were detected in WT, $App^{NL-G-F}$ (14-month-old, male), and CAPON-deficient (3-month-old, male; negative control) mice. Values shown in the graph are the fluorescence intensity of the Duolink signal divided by the intensity of Hoechst staining, expressed as the mean ± SEM ($n = 3$). Scale bar: 500 μm (left) or 100 μm (right). **b** The protein level of nNOS in the hippocampi of $App^{NL-G-F}$ mice (14-month-old male) was determined. Values shown in the graph are the band intensity of nNOS divided by the intensity of β-actin, expressed as the mean relative expression level of nNOS ± SEM ($n = 3$). **c** Representative image of the cortical region of LPS- or PBS-treated WT mice (3-month-old male) immunohistochemically stained for CAPON (green), Iba1 (red; left) or nNOS (red; right). The expression level of CAPON correlates with the level of Iba1, which is activated in LPS-induced neuroinflammation. The values shown in the graph are the fluorescence intensity of CAPON with the results expressed as the mean relative levels of CAPON ± SEM. ($n = 3$, **$p < 0.01$). Scale bar: 50 μm. Source data are provided as a Source Data file

the hippocampus (Fig. 4f). These results indicate that CAPON expression induces caspase 3-mediated neuronal cell death and results in neuronal loss and hippocampal atrophy. We also detected Gasdermin D (GSDMD) and Gasdermin E (GSDME) activation in CAPON-overexpressing mouse brains (Fig. 4g). Gasdermin family proteins possess novel membrane pore-forming activity[22]. Especially, GSDMD acts as an essential effector of pyroptosis, an inflammatory form of cell death induced by various infectious and immunological challenges[22]. This indicates that the pyroptotic pathway was also activated by CAPON overexpression. Consistently, we observed significant increases in Iba1 and GFAP signals associated with neuroinflammation in the mouse brains treated with AAV-CAPON (Fig. 4h, i). In contrast, GSDME is activated through cleavage by caspase 3 in the apoptotic pathway and induces necrosis and

pyroptosis secondarily when apoptotic cells are not scavenged[23]. Taken together, our results suggest that CAPON overexpression-mediated cell death is not caused only by a single characteristic pathway such as apoptosis, pyroptosis and necrosis but also by more complicated mechanisms that involve multiple pathways.

Notably, we observed almost identical pathological changes (neuron loss and activation of caspase 3) in single $App^{NL-G-F}$ KI and WT mice treated with AAV-CAPON (Fig. 5a). We then short-term overexpression of CAPON in hippocampus by direct injection of AAV-CAPON to single hTau-KI mice and double-KI mice (Fig. 5b). Hippocampal overexpression of CAPON also induced neuronal cell death and neuroinflammation in hTau single-KI mice as well as double-KI mice (Fig. 5c, d). These results suggest that CAPON-induced neuronal cell death does not depend on the humanization of tau or on Aβ pathology. Together

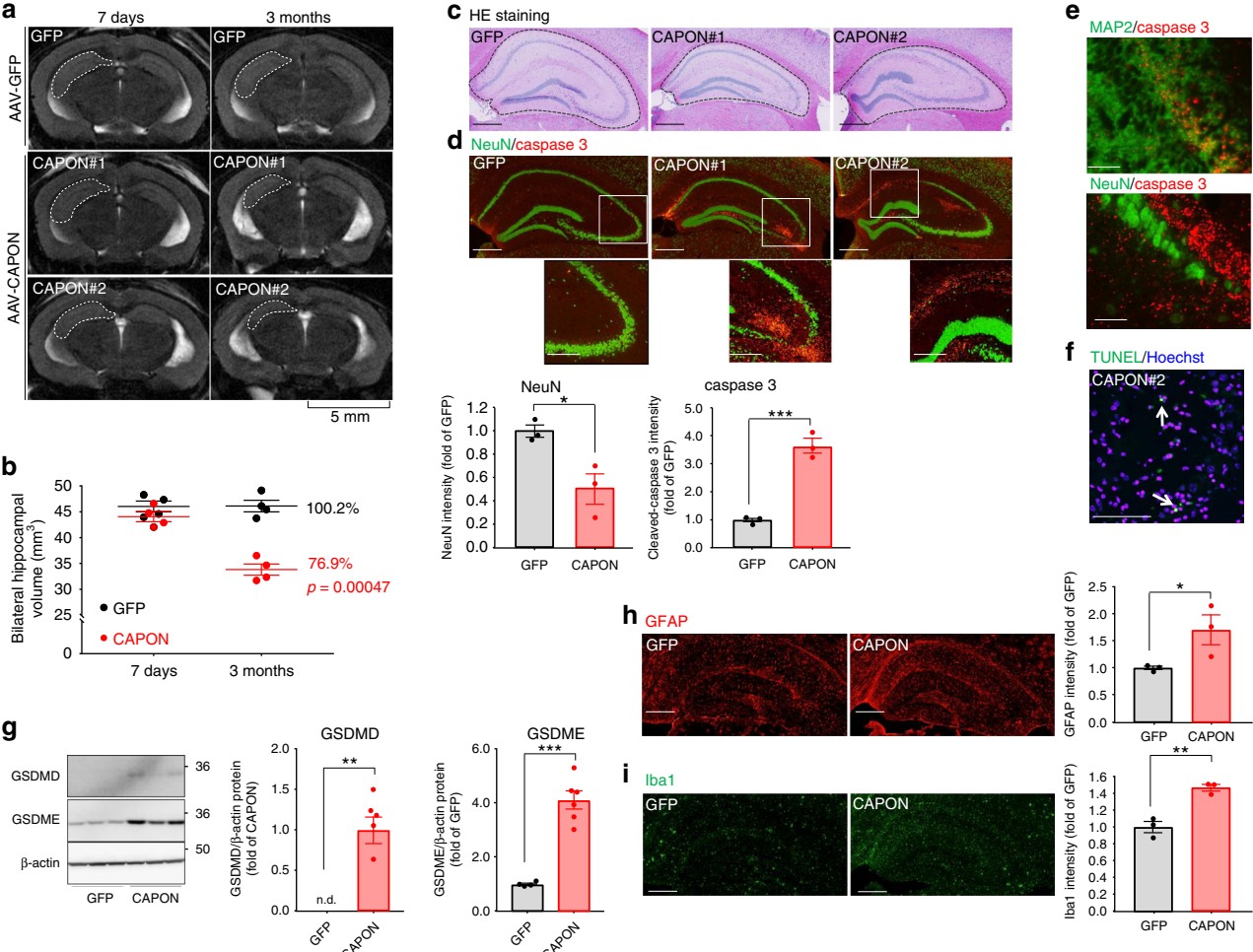

**Fig. 4** Neurodegeneration in CAPON-overexpressing $App^{NL-G-F}$/hTau double-KI mice. **a** Representative MRI scans of GFP-expressing and CAPON-overexpressing mouse brains scanned 7 days and 3 months after AAV introduction to 12-month-old double-KI mice. Scale bar: 5 mm. **b** Values shown in the graph represent the bilateral hippocampal volume calculated from 8 scanned MR images. The mean ± SEM ($n = 4$, $p = 0.00047$) is provided. **c**, **d** The hippocampal area stained by H&E (**c**) or immunostained for NeuN (green)/cleaved-caspase 3 (red) (**d**). The values shown in the graph are the fluorescence intensity of NeuN and caspase 3 with the results expressed as the mean relative levels ± SEM. ($n = 3$, *$p < 0.05$, ***$p < 0.001$). Scale bar: 500 μm (**c**, **d** above) or 250 μm (**d** below). **e** Representative image of hippocampal CA1 region of control and AAV-CAPON introduced mice co-immunostained with cleaved-caspase 3 and neuronal cell markers. Signals of cleaved-caspase 3 (red) overlap with MAP2 or NeuN signals (Green). Scale bar: 100 μm. **f** The hippocampal CA1 area of CAPON-overexpressing mice stained by TUNEL (green). Scale bar: 100 μm. **g** The protein levels of cleaved-GSDMD and cleaved-GSDME was determined. Values shown in the graph are the band intensity divided by the intensity of β-actin, expressed as the mean relative expression levels ± SEM (GFP: $n = 3$, CAPON: $n = 6$, **$p < 0.01$, ***$p < 0.001$). **h**, **i** Brain sections from GFP-expressing or CAPON-overexpressing mice immunostained with an astrocyte marker (GFAP: **h**) or a microglial marker (Iba1: **i**). The values shown in the graph are the fluorescence intensity of GFAP or Iba1 with the results expressed as the mean relative levels ± SEM. ($n = 3$, *$p < 0.05$, **$p < 0.01$). Scale bar: 500 μm. Source data are provided as a Source Data file

with the results shown in Fig. 2, this leads us to speculate that this action of CAPON occurs downstream of Aβ pathology, a notion that is supported by the fact that CAPON expression had no effect on the Aβ pathology of double-KI and $App^{NL-G-F}$ single-KI mice (Supplementary Fig. 9a, b).

**Mitochondrial damage is involved in CAPON-induced neuronal cell death.** Several studies have demonstrated that CAPON mediates signaling from the NMDA receptor in association with nNOS[9]. NMDA receptor-induced cytotoxicity is closely related to mitochondrial dysfunction because mitochondrial functions are disturbed by disruption of calcium homeostasis[24,25]. Mitochondrial dysfunction is also considered to be related to amyloid and tau pathology-induced cytotoxicity, and is detected in post-mortem AD brain and AD mouse models[26]. We therefore examined mitochondrial dysfunction in CAPON-overexpressing

mice. Upon mitochondrial damage, cytochrome C (CytC) is released into the cytosol from the mitochondria. Immunohistochemistry revealed that CAPON overexpression increases the CytC levels in the hippocampal pyramidal cell layer (Fig. 6a, b). Moreover, the protein level of Bax, which is induced by mitochondrial damage and induces neuronal cell death, was also elevated in AAV-CAPON-injected mice (Fig. 6e). We also analyzed the mitochondrial membrane potential and oxygen consumption rate of CAPON-overexpressing primary neurons: CAPAN overexpression resulted in a decrease these cellular fundamental functions compared to those of the controls (Fig. 6c, d). These results demonstrate that mitochondrial dysfunction contributes to CAPON-induced neuronal cell death.

Dexras1, a Ras family protein, is known to act as a CAPON-binding protein, and its activation inhibits Raf, MEK and ERK signaling[27]. On the other hand, activation of MEK-ERK signaling

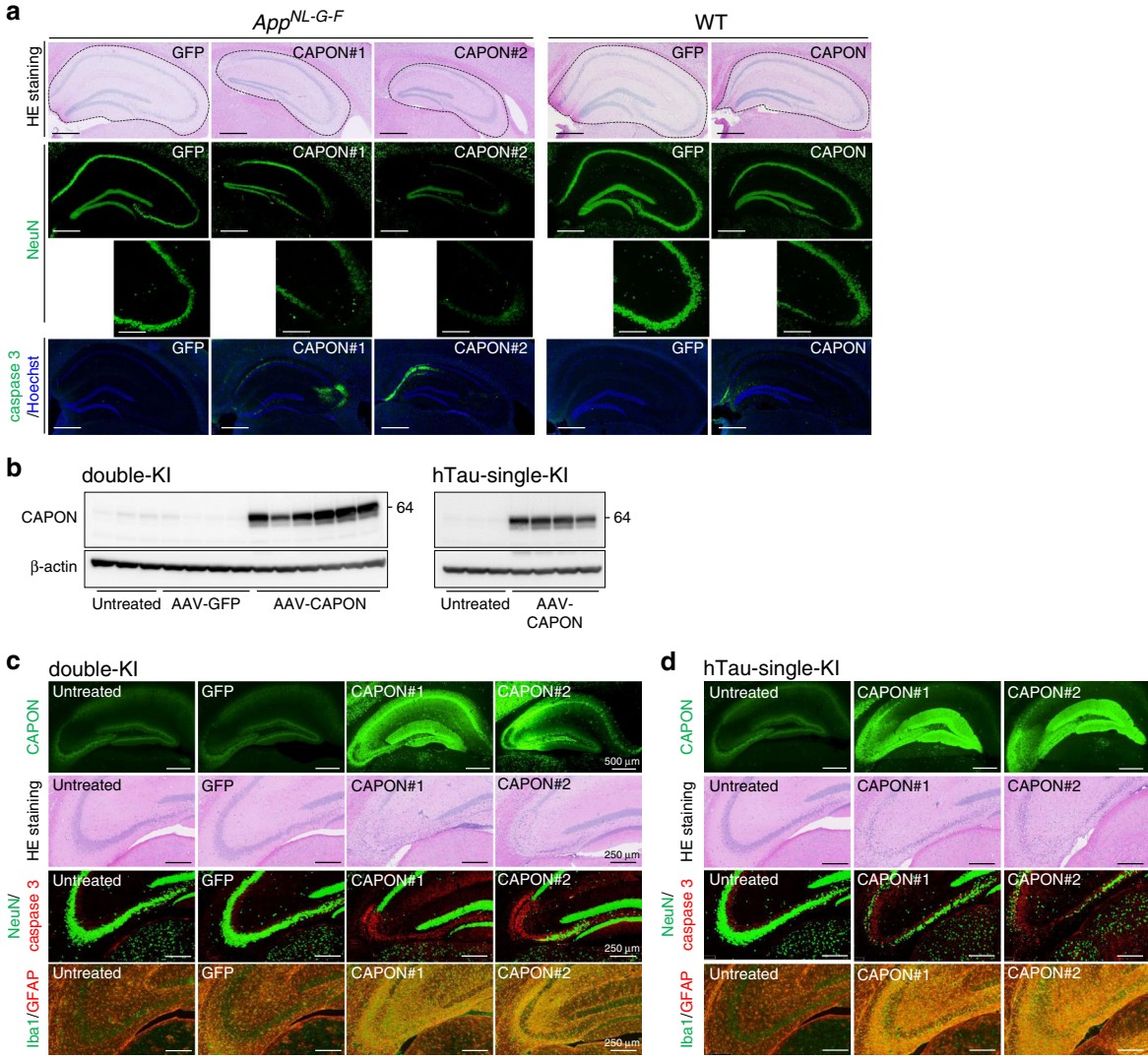

**Fig. 5** CAPON overexpression in single *App^NL-G-F* KI, wild-type or hTau-single-KI mice induces neuronal cell death. **a** AAV-GFP or AAV-CAPON was introduced bilaterally into the ventricles of 6-month-old *App^NL-G-F* (left) and WT (right) male mice. Three months later, the mice were sacrificed and paraffin sections of their brains were prepared. The brain sections were subjected to H&E staining (upper) and immunostained using NeuN (middle) or cleaved-caspase 3 (lower) antibody, respectively. CAPON overexpression enhanced hippocampal atrophy, neuron loss, and caspase 3 activation in both *App^NL-G-F* single-KI and WT mice, as well as in the double-KI mouse. Scale bar: 500 μm or 250 μm (NeuN below). **b** AAV-GFP or AAV-CAPON was introduced bilateral hippocampi of 3–4-month-old double-KI (male) and hTau-single-KI (mixture of male and female). **c**, **d** Three weeks later, the mice were sacrificed and paraffin sections of their brains were prepared. Overexpression of CAPON was determined by immunostaining (the first row). The brain sections were subjected to H&E staining (second row) and immunostained using NeuN (Green)/caspase 3 (Red) (third row) or Iba1 (Green)/GFAP (Red) (fourth row) antibody, respectively. CAPON overexpression enhanced neuron loss, and caspase 3 activation in hTau-single-KI, as well as in the double-KI. Scale bar: 500 μm (CAPON) or 250 μm (HE, NeuN/caspase 3, and Iba1/GFAP). Source data are provided as a Source Data file

has been implicated to mitochondrial damage[28,29]. We therefore examined the levels of Dexras1 and MEK-ERK phosphorylation. Our results revealed a marked decline in Dexras1 and significant acceleration of MEK-ERK phosphorylation in CAPON-overexpressing mice, suggesting that the decline in Dexras1 activates MEK-ERK signaling (Fig. 6e). Taken together, these findings indicate that CAPON could induce mitochondrial damage and neuronal cell death by regulating Dextras1-MEK-ERK signaling (Fig. 6e). Zhu et al suggested that overexpression of CAPON activates Dexras1 activity by facilitating the s-nitrosylation of Dexras1[27]. They detected Dexras1 activation after several days of lentivirus-mediated overexpression of CAPON in the mouse brain. In contrast, we observed a decrease in Dexras1 after 3 months, suggesting that prolonged overexpression of CAPON decreases the level of the Dexras1

protein via negative feedback, and results in hyperactivation of the MEK-ERK signal. Incidentally, the baseline expressions of Dextras1 and phosphorylated (p)-MEK were essentially identical between WT and double-KI mice, and we also observed alterations of these proteins in CAPON-overexpressing WT mice (Supplementary Fig. 10).

**CAPON expression facilitates tau phosphorylation and insolubility but not amyloid pathology.** As we originally identified CAPON as a tau-binding protein, the interaction between CAPON and tau could be considered a cause of tau pathology, resulting in the induction of neuronal cell death. We therefore tested whether CAPON overexpression aggravates tau pathology (Fig. 7). We first validated the interaction between tau and CAPON in the WT, hTau-KI, *App^NL-G-F*, and double-KI mouse

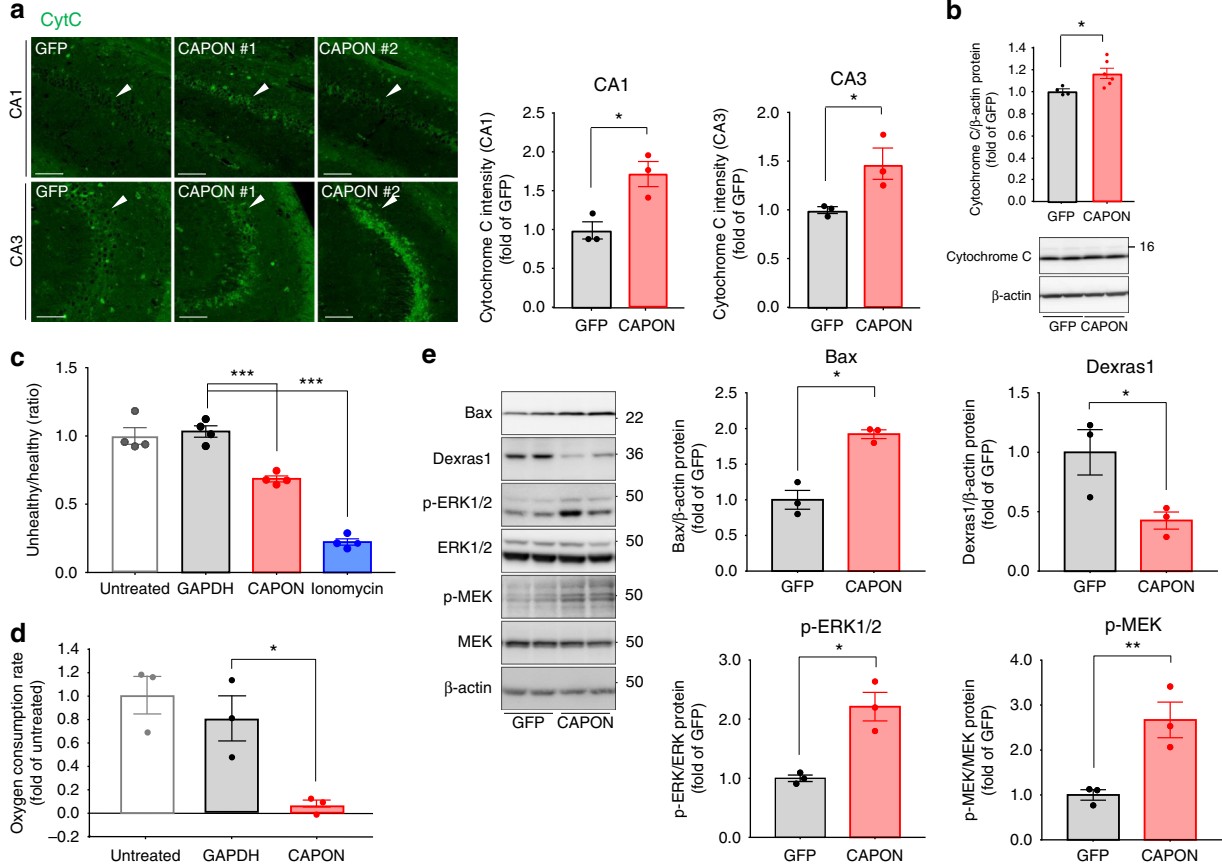

**Fig. 6** CAPON induces mitochondrial damage via Dexras1/MEK/ERK signaling. **a** AAV was introduced into 12-month-old male double-KI, and brain samples were prepared 3 months after injection. Hippocampal CA1 or CA3 region of GFP-expressing or CAPON-overexpressing double-KI mice immunostained with CytC. The values shown in the graph are the fluorescence intensity of CytC in CA1 and CA3 region with the results expressed as the mean relative levels ± SEM. ($n = 3$, *$p < 0.05$). Scale bar: 100 μm. **b** The protein level of Cyrochrome C in the cytosolic fraction of hippocampi. Values shown in the graph are the band intensity divided by the intensity of β-actin, expressed as the mean relative expression levels ± SEM (GFP: $n = 3$, CAPON: $n = 6$, *$p < 0.05$). **c** CAPON or GAPDH (Control) was overexpressed in primary cortical neurons from WT mice and mitochondrial membrane potential was detected using the JC-1 Mitochondrial Membrane Potential Assay. Ionomycine was treated to obtain a membrane depolarization positive control. Values show the fluorescence ratio for healthy membrane potential to unhealthy membrane potential ($n = 4$, ***$p < 0.001$). **d** Oxygen consumption rate was measured using CAPON or GAPDH (Control) was overexpressed in primary cortical neurons from WT by Oxygen Consumption Rate Assay Kit. Values show the relative oxygen consumption rate which are calculated from change of fluorescence intensity per 1 min while the fluorescence rise linearly. **e** Levels of Dexras1/MEK/ERK signaling-related proteins were detected using hippocampal samples from GFP-expressing or CAPON-overexpressing double-KI mice. The values shown in the graph are the band intensity of proteins divided by the intensity of β-actin, with the results expressed as the mean level ± SEM ($n = 3$, *$p < 0.05$, **$p < 0.01$). Source data are provided as a Source Data file

brain using the Duolink system (Fig. 7a and Supplementary Fig. 11a). We then assessed pathological changes in tau in CAPON-overexpressing double-KI mice (Fig. 7b–f). Intriguingly, CAPON overexpression significantly increased the phosphorylation level at all the phosphorylation sites which we tested, without any change in the total tau level (Fig. 7b). To investigate the formation of insoluble inclusions such as NFTs in CAPON-overexpressing mice, we isolated a 1% sarkosyl-insoluble fraction of hippocampal protein and detected insoluble tau by immunoblotting using the Tau5 antibody. A greater amount of sarkosyl-insoluble Tau was detected in CAPON-overexpressing mice compared to control mice (Fig. 7c). We also observed a higher amount of soluble tau oligomer, which could be detected by non-reducing SDS-PAGE, in the CAPON-overexpressing mice (Fig. 7d). An increase in AT8-positive tau was also seen by immunohistochemistry not only around Aβ plaques but also over a wide area of the hippocampus (Fig. 7e). In addition, we found that the MC1 antibody, which recognizes the pathological conformation of tau protein, detected some neuronal cells in CA1

pyramidal cell layer (Fig. 7f). Consistently, these cells were also stained for phosphorylated-Ser404 (pS404) rabbit antibody (Fig. 7f). These results indicate that CAPON can drive the progression of tau pathology in the mouse brain. We also observed the induction of tau pathology by AAV-CAPON not only in double-KI mice but also in $App^{NL-G-F}$single-KI and WT mice (Supplementary Fig. 11b, c), demonstrating that amyloid deposition or the humanization of tau did not affect the CAPON-induced tau pathology.

How does CAPON facilitate the phosphorylation and accumulation of tau? Many reports indicate that CAPON essentially functions as a scaffold protein for nNOS. We therefore speculated that it induces the nitration of tau, which is associated with the progression of tauopathy. To investigate this, we analyzed nitration at the 18[th] tyrosine (Y18) and Y29 of tau using nitrated-tyrosine antibodies, revealing significant enhancement of nitration at Y29 in CAPON-overexpressing mice (Fig. 7g). This suggests that nitration at Y29 is enhanced by CAPON, resulting in pathological tau aggregation. Indeed, Reynolds et al, have

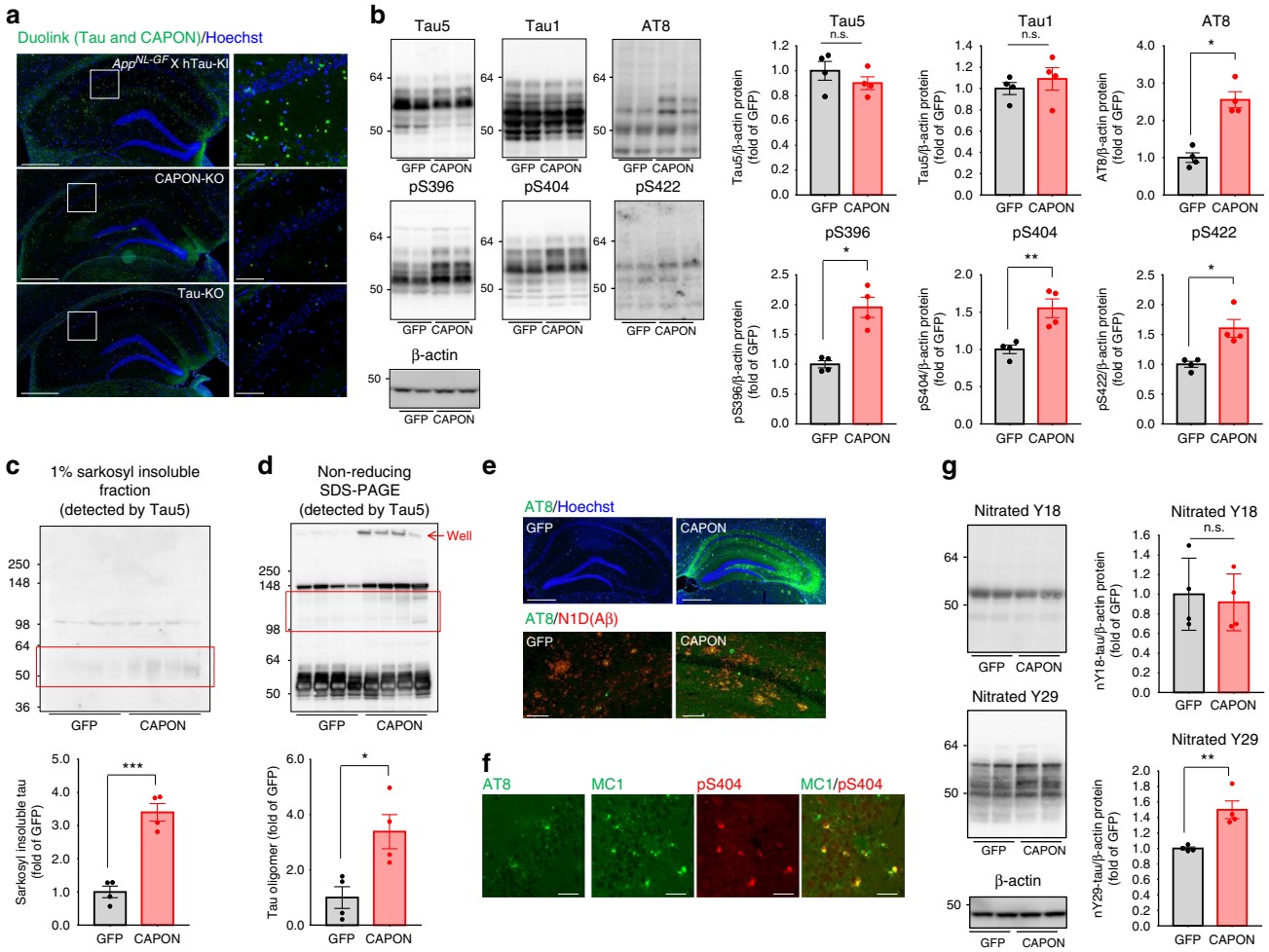

**Fig. 7** Tau pathology in the CAPON-overexpressing $App^{NL\text{-}G\text{-}F}$/hTau double-KI mouse. **a** Duolink signals (green) generated by tau-CAPON interaction were detected in $App^{NL\text{-}G\text{-}F}$/hTau double-KI, Tau-KO and CAPON-deficient (negative control) mice. Scale bar: 500 μm (left) or 100 μm (right). **b** AAV was introduced into 12-month-old male double-KI, and brain samples were prepared 3 months after injection. The total-tau and phosphorylated-tau levels in the soluble (S1) fraction from GFP-expressing or CAPON-overexpressing double-KI mice were determined using several tau antibodies. The values shown in the graphs are the band intensity of the proteins divided by the intensity of β-actin, with the results expressed as the mean ± SEM ($n = 4$, *$p < 0.05$, **$p < 0.01$). **c** The sarkosyl-insoluble (P3) fraction was obtained from the hippocampus, and insoluble tau levels were determined using Tau5 antibody ($n = 4$). **d** Tau oligomers in the S1 fraction were detected by non-reducing SDS-PAGE and immunoblotting with Tau5 antibody ($n = 4$, ***$p < 0.001$). The red box shows bands which appear to be tau oligomer ($n = 4$, *$p < 0.05$). **e** AT8-positive tau was detected by immunohistochemistry. The lower panels show the hippocampal CA1 region double-stained with AT8 and N1D (Aβ). Scale bar: 500 μm (above) or 100 μm (below). **f** Magnified image of CAPON-overexpressing double-KI mice immunostained with AT8, MC1 or pS404-tau antibody. The images show hippocampal CA1 region. Scale bar: 25 μm. **g** Nitrated-tau levels in the S1 fraction were determined using anti-nitrated-Y18 (nY18) and nY29 antibodies. The values shown in the graph are the band intensity of proteins divided by the intensity of β-actin, with the results expressed as the mean ± SEM ($n = 4$, **$p < 0.01$). Source data are provided as a Source Data file

reported that tau with nitrated Y29 is more abundant in NFTs than soluble tau, with the authors suggesting that nitration at Y29 is a disease-related event that may alter the intrinsic ability of tau polymerization[30].

We next questioned which event is first initiated in CAPON-overexpressing mice, tau pathology or neurodegeneration. To answer this question, we analyzed the pathology of AAV-CAPON-expressing WT mice in which overexpression of CAPON occurs at a low level (Fig. 8a). Low level overexpression of CAPON did not induce hippocampal atrophy, neuron loss, or caspase 3 activation in double-KI mice (Fig. 8b–d). On the other hand, these mice exhibited a significant increase in phosphorylated tau (Fig. 8e). Moreover, we observed a significant increase in AT8-positive tau without neuronal cell death 2 weeks after the introduction of AAV-CAPON (Fig. 8f–h). These results suggest that tau pathology is induced by a lower level of CAPON, whereas

neuronal cell death needs a higher level or more prolonged action of the protein.

**CAPON induces neuronal cell death in tau-KO mice.** From the above results, it appears that tau pathology precedes neuronal cell death in CAPON-overexpressing mice. This conclusion led us to investigate whether CAPON-induced neuronal cell death is mediated by tau pathology. To this end, we assessed neuronal cell death and brain atrophy in AAV-CAPON-expressing tau-KO mice (Fig. 9). We injected AAV-CAPON into 6-month-old tau-KO or WT mice (Supplementary Fig. 12), and assessed their brain pathologies 3 months later. Interestingly, much higher levels of CAPON were expressed in the tau-KO mice, although the reason for this remains unknown (Fig. 9a). MRI and histochemical analyses revealed that the degree of hippocampal atrophy

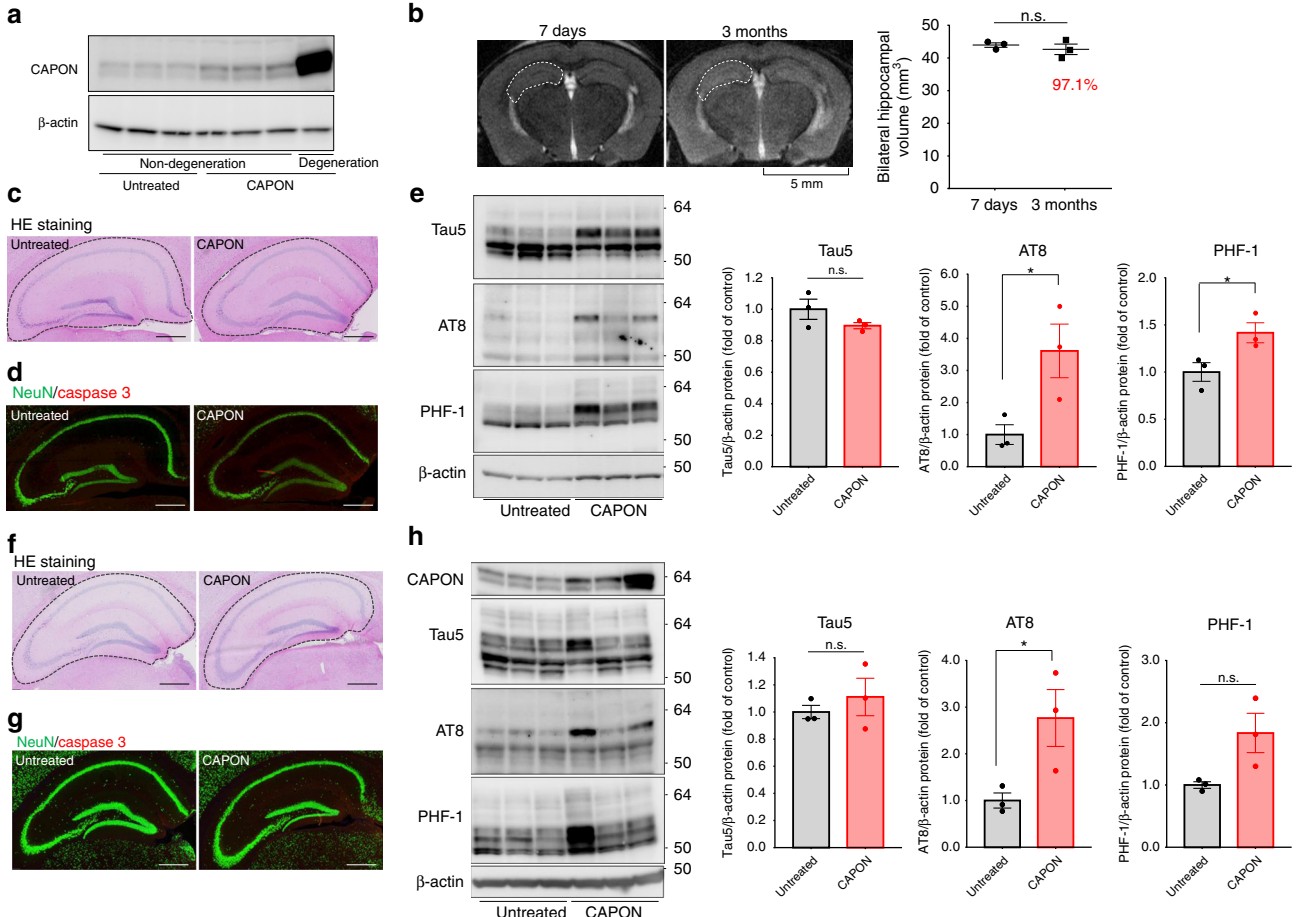

**Fig. 8** A low level of CAPON overexpression induces tau phosphorylation without neuronal cell death. **a–e** AAV-CAPON was introduced into 6-month-old WT male mice. Data were obtained 3 months after AAV injection. **a** Western blots show the overexpression levels of CAPON in the hippocampus of mice with and without neurodegeneration. **b** Representative MRI scan of the low-level-CAPON-overexpressing mouse brain, which does not show neuronal cell death. Scale bar: 5 mm. **c, d** Hippocampal area of low-level-CAPON-overexpressing mice stained by H&E (**c**) or immunostained for NeuN (green)/cleaved-caspase 3 (red) (**d**). Scale bar: 500 μm. **e** The levels of total tau and phosphorylated tau in low-level-CAPON-overexpressing mice were determined using several tau antibodies. Values shown in the graph represent the band intensity of proteins divided by the intensity of β-actin, with the results expressed as the mean ± SEM ($n = 3$, *$p < 0.05$). **f–h** AAV-CAPON was introduced into 3-month-old WT male mice. Data were obtained 2 weeks after AAV-injection. **f, g** Hippocampal area of CAPON-overexpressing mice stained by H&E (**f**) or immunostained for NeuN (green)/cleaved-caspase 3 (red) (**g**). Scale bar: 500 μm. **h** The levels of CAPON, total tau and phosphorylated tau were determined in CAPON-overexpressing mice. The values shown in the graph represent the band intensity of proteins divided by the intensity of β-actin, with the results expressed as the mean ± SEM ($n = 3$, *$p < 0.05$). Source data are provided as a Source Data file

corresponded with the expression level of CAPON (Fig. 9a–d and Supplementary Fig.13). Moreover, we observed significant activation of caspase 3 and neuronal loss in the AAV-CAPON-expressing tau-KO mice (Fig. 9e). These results indicate that CAPON induces neuronal cell death via a tau-independent mechanism or through a combination of both tau-dependent and tau-independent mechanisms. We would also like to point out that tau is not the only microtubule-binding protein (MAP) and that the other MAPs such as MAP2 might be involved although we do not have any experimental evidence[31,32].

**Deficiency of CAPON restores AD-related pathological phenotypes in P301S-Tau-Tg mice.** We thus far analyzed the effect of CAPON overexpression on pathological characteristics in double-KI mice. As a proof of evidence, we next examined the effect of CAPON deficiency on tau pathology and neurodegeneration in tauopathy mouse model. The P301S-Tau-Tg mice on the CAPON KO background[33] exhibited significantly reduced levels of total and phosphorylated tau (Fig. 10a). CAPON deficiency also resulted in reduction of Sarkosyl-insoluble tau in

P301S-Tau-Tg mice (Fig. 10b), suggesting that CAPON contributes to progression of tau pathology in P301S-Tau-Tg mice. Moreover, CAPON deficiency ameliorated brain atrophy and neuron loss in P301S-Tau-Tg mice (Fig. 10c, d). These results suggest that the CAPON-induced neuronal cell death is closely associated with the pathological tau protein although there appears to be a tau-independent mechanism as well (Fig. 9a).

**Discussion**

In this study, we have demonstrated that Aβ pathology leads to the accumulation of CAPON protein, and that the increase in CAPON induces tau pathology and neuronal cell death. Our findings suggest that CAPON is one of the novel mediators that link Aβ, tau and neurodegeneration. CAPON was originally identified as a nNOS-binding protein and is considered to be involved in NMDA receptor-mediated excitotoxicity[9]. Although one report proposed that CAPON prevents excitotoxicity by competitively inhibiting nNOS-PSD95 binding[9], recent studies have indicated that it mediates NMDA receptor-induced neuronal toxicity[13,34,35]. Our study supports the latter proposal, given

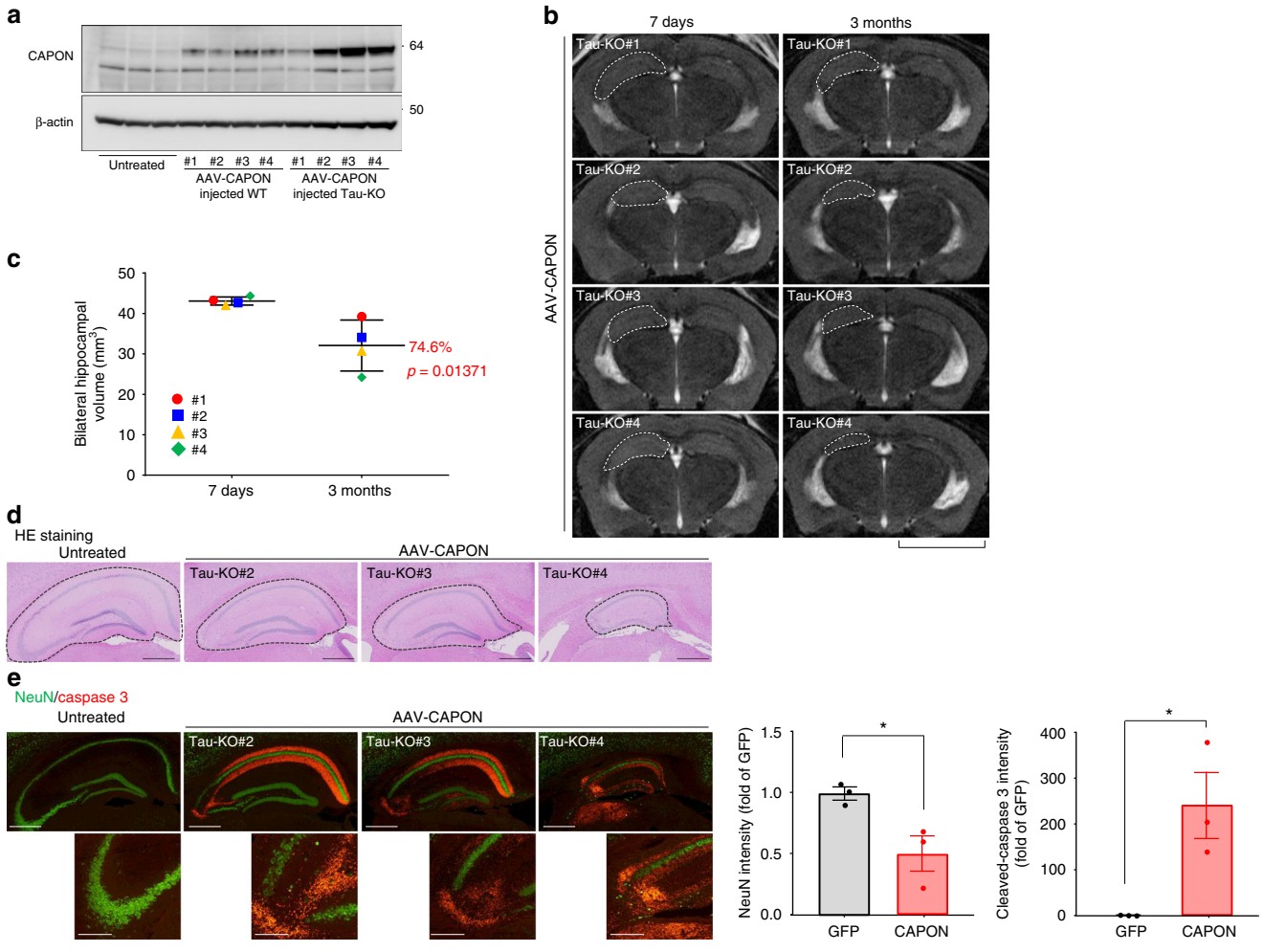

**Fig. 9** CAPON overexpression in tau-KO mice induces neurodegeneration. **a** AAV-CAPON was introduced into 6-month-old WT (male) or tau-KO mice (#1, #2 and #4 are male, and #3 is female), and expression levels of CAPON in the hippocampus were determined 3 months after injection. **b** Representative MRI scan of the CAPON-overexpressing tau-KO mouse brain scanned 7 days and 3 months after the introduction of AAV. Scale bar: 5 mm. **c** The values shown in the graph are the bilateral hippocampal volume calculated from 8 scanned MR images ($n = 4$, $p = 0.01371$). **d, e** The hippocampal area of CAPON-overexpressing tau-KO mice stained by H&E (**d**) or immunostained for NeuN (green)/cleaved-caspase 3 (red) (**e**). Scale bar: 500 μm (**d, e** above) or 100 μm (**e** below). Source data are provided as a Source Data file

our observation that overexpression of CAPON leads to neuronal cell death and hippocampal atrophy. We also demonstrated the contribution of mitochondrial dysfunction to this pathology. Aβ oligomers have been reported to upregulate reactive oxygen/nitrogen species (ROS/RNS) and induce mitochondrial dysfunction via their interaction with the NMDA receptor[36]. Abnormal activation of nNOS signaling together with mitochondrial $Ca^{2+}$ overload produces excess ROS and RNS[37]. This in turn leads to abnormal s-nitrosylation, sulfonation and the accumulation of peroxides, resulting in protein misfolding and dysfunction[37–39]. For example, s-nitrosylation of dynamin-related protein 1 (Drp1) is linked to mitochondrial dysfunction and neuronal injury, and protein disulfide isomerase is associated with protein misfolding in neurodegenerative disease[40,41]. Dysfunction of CAPON may contribute to disruption of these proteins, thereby producing pathological changes. Mitochondrial dysfunction also induces caspase activation and cell death[42].

In CAPON-overexpressing mice, we detected not only apoptosis markers such as cleaved-caspase 3 but also non-apoptotic cell death markers, i.e. cleaved-GSDMD and GSDME. GSDMD is a physiological substrate for inflammatory caspases, and its activation is required for initiation of pyroptosis. GSDME activation by caspase 3 induces pyroptosis and necrosis secondarily if

apoptotic cells are not scavenged[23]. Our observation suggests that CAPON-induced neuronal cell death is not composed of a single simple pathway but also of complicated mechanisms, which involve multiple pathways. Indeed, activation of several cell death pathways were also detected in AD samples and AD mouse models. For instance, NLRP1-dependent neuronal pyroptosis is detected in APPswe/PS1dE9[43]. Also, activation of necroptosis pathway, a programmed form of necrosis, is observed in AD samples[44]. Taken together, the mechanisms of neurodegeneration observed in the present study appear quite complex, and we thus need further investigation to understand the entire landscape while taking contribution of neuroinflammation into consideration. It is possible that neuroinflammation may be attributed as a cause of cell death induced by CAPON overexpression.

We observed a significant decrease in Dexras1 and enhanced MEK-ERK phosphorylation in CAPON-overexpressing mice. Activation of MEK-ERK signaling is related to the induction of mitochondrial damage[28,29]. We therefore concluded that Dexras1-MEK-ERK signaling is one of the mediators of CAPON-induced neuronal cell death. In contrast, Chen et al[45] demonstrated that a deficiency in Dexras1 inhibits NMDA-induced neuronal cell death. Moreover, Zhu et al[27] reported that CAPON overexpression facilitates Dexras1 activity via enhancement of its

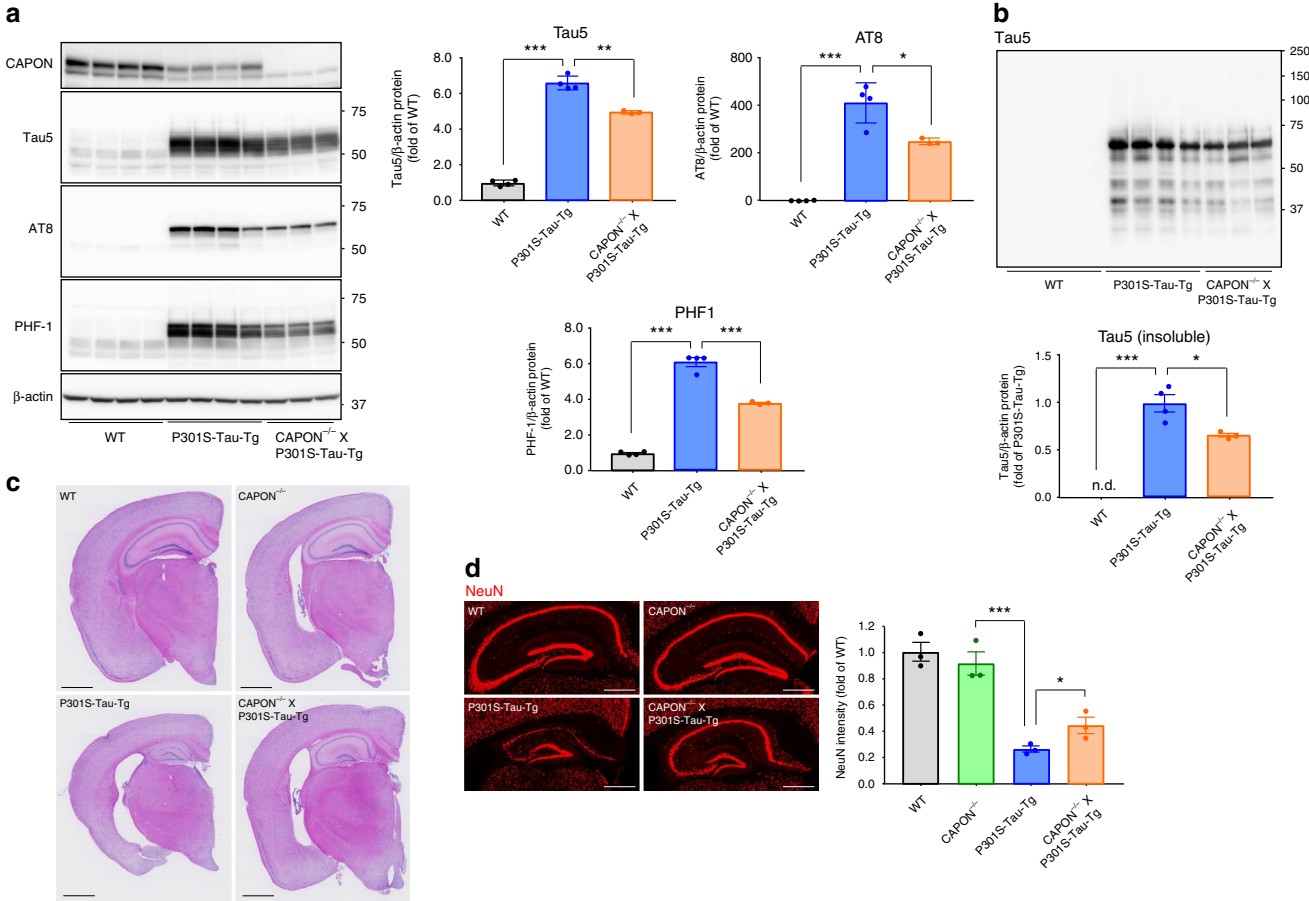

**Fig. 10** CAPON deficiency in P301S-Tau-Tg mice attenuates tau pathology and neurodegeneration. **a** The total-tau and AT8/PHF1 positive-tau levels in the soluble (S1) fraction from 12-month-old WT, P301S-Tau-Tg or CAPON$^{-/-}$/P301S-Tau-Tg mice (mixed-sex) were determined. Values are the band intensity of the Tau5/AT8/PHF1 divided by the intensity of β-actin, with the results expressed as the mean ± SEM (WT, P301S-Tau-Tg: n = 4, CAPON$^{-/-}$/P301S-Tau-Tg:n = 3, *p < 0.05, **p < 0.01, ***p < 0.001). **b** The total-tau sarkosyl insoluble fraction were determined. Values are the band intensity of the Tau5, with the results expressed as the mean ± SEM (WT, P301S-Tau-Tg: n = 4, CAPON$^{-/-}$/P301S-Tau-Tg:n = 3, *p < 0.05, ***p < 0.001). **c, d** The sections of WT, CAPON$^{-/-}$, P301S-Tau-Tg or CAPON$^{-/-}$/P301S-Tau-Tg mice stained by H&E (**c**) or immunostained for NeuN (**d**). The values shown in the graph are fluoresence intensity of NeuN, with the results expressed as the mean level ± SEM (WT, P301S-Tau-Tg: n = 4, CAPON$^{-/-}$/P301S-Tau-Tg:n = 3, *p < 0.05, ***p < 0.001). Scale bar: 1 mm (**c**) or 500 μm (**d**). Source data are provided as a Source Data file

s-nitrosylation, and that this inactivates MEK-ERK signaling. We assumed that this inconsistency is due to the difference in experimental conditions between studies, as Zhu et al,[27] used a lenti-virus to overexpress CAPON, observing activation of Dexras1 and inactivation of MEK-ERK signaling 4 days after introduction of the virus. We also assumed that a reduction in activated Dexras1 rather than a deficit in Dexras1 *per se* facilitated MEK-ERK signaling, and disrupted mitochondrial function. This effect could be mediated by prolonged overexpression of CAPON.

Although we could detect a significant increase in CAPON protein in the hippocampus and cortex of the *App*$^{NL-G-F}$ mouse, these mice do not exhibit severe neuronal cell death or tau pathology[5]. In sporadic AD patients, Aβ accumulation starts about 20 years prior to the clinical onset of the disease. We therefore assume that the prolonged effect of Aβ leads to the accumulation of CAPON and results in tau pathology and neuronal cell death.

We found that the nitration of tyrosine at Y29 is one of the mechanisms by which tau pathology is induced. Nitrated-Y29-positive tau is predominately detected in tau inclusions such as NFTs, suggesting that nitration at Y29 facilitates the aggregation of tau[30]. We attempted to validate the contribution of nNOS to tau nitration using nNOS-deficient mice that overexpress

CAPON. However, we were unable to successfully overexpress CAPON by AAV delivery in nNOS-KO mice, probably because the protein is not stable in these animals. CAPON overexpression-induced neuroinflammation may also contribute to acceleration of tau pathology through other possible mechanisms. For instance, genome-wide and rare variant association studies have shown linkage between single-nucleotide polymorphisms in genes related to immune system and etiology of AD. In particular, Triggering receptor expressed on myeloid cells 2 (TREM2), the mutation in which increases the risk of AD, is associated with tau pathology and synaptic loss in post mortem human cortical samples from AD patients[46]. Also, Asai et al[47] demonstrated that depletion of microglia and inhibition of exosome synthesis interrupted tau propagation, suggesting that microglia and exosomes contribute to the progression of tauopathy. Thus, although there is still room for argument about relationship between neuroinflammation and tau pathology, CAPON may be a candidate facilitater of tau pathology in such processes.

Given that tau pathology occurred earlier than neurodegeneration in CAPON-overexpressing mice, we predicted that this pathology was driving neurodegeneration. However, we observed CAPON-induced hippocampal atrophy and neuronal cell death

not only in double-KI mice but also in tau-KO mice. This result indicates that CAPON-induced neuronal cell death occurs via a tau pathology-independent manner or through a combination of both tau-dependent and tau-independent mechanisms. Since CAPON deficiency in tauopathy mouse model attenuated tau accumulation and neuron loss, the CAPON-induced cell death can be attributed at least in part to a tau-dependent mechanism. A positional correlation between tau pathology and neuron loss is observed in the human AD brain, while it remains unclear whether tau is the actual killer of neuronal cells. Although overexpression of tauopathy-associated mutant tau induces neuronal cell death, overexpression of WT tau or artificial facilitation of tau propagation by aggregate seeding almost never induces such death. Hence, the relationship between tau pathology and neurodegeneration still requires further investigation.

Numerous studies have reported that CAPON is a risk gene for schizophrenia and other psychiatric disorders, including autism spectrum disorder, obsessive-compulsive disorder, post-traumatic stress disorder and depression[10,48–50]. Psychiatric symptoms are a major feature of AD, and sometimes appear from the early stages of the disease[51,52]. The disruption of CAPON expression induced by Aβ may therefore link these clinical conditions.

A recent study used the APP/PS1 mouse and an in vitro system to demonstrate that the interaction between CAPON with nNOS is enhanced under amyloid pathologies, and that this facilitates neuronal toxicity[53]. In contrast, we did not detect an enhanced interaction between CAPON and nNOS in App KI mice. However, because these mice, unlike APP/PS1 animals, do not present significant neuron loss or neuronal cell death, it is possible that such an effect might not be detected. We assume that the gradual accumulation of CAPON over long periods could enhance the CAPON-nNOS interaction and induce neuronal toxicity, particularly as Aβ accumulation starts more than two decades before the onset of the clinical phenotype. Another reason for the difference between App KI and APP/PS1 mice could lie in the overexpression of APP and PS1 in the latter case. We have previously suggested that APP and/or presenilin overexpression itself alters calcium homeostasis without regard for amyloid pathology[54–57]. For example, unlike App KI mice, APP-overexpressing mice display calpain-dependent conversion of p35 to p25, which upregulates the kinase activity of CDK5[56]. Moreover, APP/PS1 mice exhibit enhanced endoplasmic reticulum stress, whereas App KI and APP-single transgenic mice do not[55,57]. CAPON and nNOS function could therefore be subjected to the action of calcium signaling, and care is required when interpreting results obtained using the APP/PS1 model.

It is desirable to prevent the progression of AD before the onset of neurodegeneration. Many groups have been developing novel therapeutic agents to target Aβ and tau accumulation[58]. However, few studies are targeting the connection between amyloid pathology and tau pathology or neuronal cell death. We believe that CAPON is a promising drug target to break the connection between Aβ and tau or cell death. Inhibition of the interaction between CAPON-tau or CAPON-nNOS could therefore be a novel approach for the treatment of AD and related diseases.

## Methods

**Animals**. All animal experiments were conducted in accordance with the guidelines of the RIKEN Center for Brain Science. We previously produced the $App^{NL-G-F/NL-G-F}$ KI ($App^{NL-G-F}$) mouse using genomic DNA of introns 15 to 17 of mouse App, which humanized the Aβ sequence and introduced KM670/671NL (Swedish), I716F (Iberian), and E693G (Arctic) mutations[5]. Human MAPT KI (hTau-KI) mice, in which the entire human Mapt gene was inserted at the murine Mapt gene locus, were generated. The strategy for generating this mouse is described in the Supplementary information (Fig. S1). The Wtau-Tg mouse expresses wild-type (WT) human 4 repeat Tau tagged with myc and Flag epitopes which are regulated by the CAM kinase II promoter[8]. The P301S-Tau-Tg

(Line PS19) mice were created on a B6C3H/F1 background[15]. For our experiments, PS19 mice were back-crossed onto a C57BL/6 background. The details of the CAPON (Nos1ap)-KO mouse have been described in Sugiyama et al[33]. Tau-KO (knockout) (B6.129×1-Mapttm1Hnd/J) and WT C57BL/6 mice were purchased from Jackson Laboratory (Bar Harbor, ME). All strains were maintained on a C57BL/6 background.

**Materials**. The cDNA encoding mouse CAPON (NCBI Reference Sequence: NM_001109985) was obtained from Origene plasmid (CAT#: MC217011) (Origene, Rockville, MD). The AAV9 vectors were produced as previously described[59]. The AAV vector plasmids contained an expression cassette, consisting of a synapsin (Syn) I promoter (GenBank, M55300.1), followed by cDNA encoding CAPON or green fluorescent protein (GFP), and a simian virus (SV) 40 polyadenylation signal sequence; and consisting of a Cytomegalovirus (CMV) promoter followed by cDNA encoding GAPDH and a SV40 polyadenylation signal sequence. The antibodies used for immunohistochemistry or western blotting are summarized in Supplementary Table 1. We confirmed the specificity of the CAPON antibodies using the CAPON-KO mouse[33] (Supplementary Fig. 3).

**Generation of human MAPT KI (hTau-KI) mice**. As described in Supplementary Figure 1a, we isolated genomic DNA of the human MAPT gene (NCBI Reference Sequence: NG_007398) with H2 haplotype, from the ATG codon of exon 1 to the 3'-untranslated region (67440 bp), from the human bacterial artificial chromosome (BAC) library. We inserted a neo gene cassette with a lox/FRT sequence at the 3'-end of the isolated human MAPT gene which was subsequently inserted into a humanized BAC vector, as a humanized BAC targeting vector.

Gene-replacement of the mouse Mapt gene with its human MAPT gene counterpart, which includes a region starting from the ATG codon of exon 1 to the 3'-untranslated region (58019 bp), was carried out with the homologous-based recombination technique using BA1 C57BL/6 × 129SvEv hybrid embryonic stem (ES) cells. Targeted ES cells were microinjected into C57BL/6 blastocysts. Resulting chimeras with a high percentage agouti coat color were mated to wild-type C57BL/6 mice to generate F1 heterozygous offspring. We extracted DNA from the biopsied tails of mouse pups and identified the F1 generation by PCR.

We then crossbred heterozygous mutant mice with EIIa-Cre Tg mice to remove the neo gene and removed the EIIa-Cre transgene by crossing the mice with wild-type C57BL/6 mice. The generated MAPT KI mice were then back-crossed to the C57BL/6 J strain. We genotyped the mice by PCR using the following primers: Fwt5: 5′-GTCAGATCACTAGACTCAGC-3′, Rwt5: 5′-CTGTGCTCCACTGTGACTGG-3′ and Rhm5: 5′-CTGCTTGAGTTATCTTGGCC-3′.

**Tau interactome**. Tau was immunoprecipitated from brain extracts of 9-month-old WT or Wtau-Tg mice with Flag-tag antibody. The whole brains except the olfactory bulb and cerebellum were extracted in extraction buffer (phosphate-buffered saline (PBS) containing, protease inhibitor cocktail, phosphatase inhibitor), and insoluble material was removed by centrifugation at $1000 \times g$ for 10 min. The resulting supernatant was adjusted to 1 mgmL⁻¹ by adding extraction buffer. Next, 1 mL of supernatant was incubated with anti-FLAG M2 magnetic beads (Sigma-Aldrich, St. Louis, MO) for 2 h at 4 °C. After incubation, the magnetic beads were collected by placing the reaction tubes in a magnet stand, then washed 5 times with PBS. The precipitated proteins were extracted with 0.1% trifluoroacetic acid (pH 2.5), and dried using a speed-vacuum system. After removal of the solvent, the samples were dissolved in 100 mM ammonium bicarbonate, 0.2% RapiGest (Waters Corporation, Milford, MA), 2 mM CaCl₂, and 100 ng μL⁻¹ trypsin, then digested by incubation at 37 °C for 48 h. The determination of peptides was carried out by the Support Unit for Bio-Material Analysis, Research Resource Division of the Center for Brain Science, using an Q Exactive Hybrid Quadrupole-Orbitrap mass spectrometer (Thermo Fisher Scientific, Waltham, MA). The peptide identification was performed using Proteome Discover (Thermo Fisher Scientific) and the Swiss-prot database. Co-immunoprecipitated proteins which were specifically identified in Wtau-Tg mice are listed in Supplementary Data 1.

**AAV injection**. WT, $App^{NL-G-F/NL-G-F}$ KI ($App^{NL-G-F}$), $App^{NL-G-F/NL-G-F}$ KI X hTau-KI (double-KI), hTau-KI and tau-KO mice (3–12 months old) were used for the experiments. AAV vectors were diluted in PBS containing 2% PEG400 to $1 \times 10^{10}$ genome copies per 5 μL or 1 μL. Mice were anesthetized with pentobarbital (50 mg kg⁻¹, i.p.) and placed in a stereotaxic apparatus before administration of the viral vector solution. AAV solution (5 μL for icv injection per 1 μL for hippocampal injection) was injected bilaterally into the cerebral ventricles (stereotaxic coordinates: anteroposterior, −0.45 mm; mediolateral, ± 1 mm; dorsoventral, −2.4 mm) or hippocampi (stereotaxic coordinates: anteroposterior, −2.7 mm; mediolateral, ± 3.1 mm; dorsoventral, −2.4 mm) using a needle equipped with a 50 mL NanoSyringe (Altair Corporation, Yokohama, Japan), at a constant flow rate of 1 μL min⁻¹ (icv) or 0.2 μL min⁻¹ (hippocampal) using a Legato 130 syringe pump (KD Scientific, Holliston, MA). After AAV injection, mice were bred in plastic cages with food (CE2, Clea Japan, Tokyo, Japan) and water, and were maintained on a 12/12 h light–dark cycle for 3 months (icv) or 3 weeks (hippocampal).

**Magnetic resonance imaging**. We conducted magnetic resonance imaging (MRI) of 21-month-old *MAPT* KI (Supplementary Fig. 2c), and AAV-CAPON/GFP-expressing mice 7 days and 3 months after AAV injection (Figs. 4a and 9b). The mice were anchored in the apparatus under anesthesia with 1.5% (v/v) isoflurane. During the scanning, the depth of anesthesia was monitored with a breathing sensor. Coronal/horizontal T2-weighted (T2W) MRI scans (2D TurboRAGE) of the whole brain were performed with a vertical-bore 9.4 T Bruker AVANCE 400WB imaging spectrometer with a 250 mTm$^{-1}$ actively shielded imaging gradient insert (Bruker BioSpin, Billerica, MA) controlled by Paravision software. T2W scans were performed with the following parameter settings: TR (repetition time) = 4342.2 ms, TE (echo time) = 53.8 ms, matrix dimensions = $256 \times 256$, flip angle = 180 degrees, field of view = $1.8$ cm $\times 1.8$ cm. We used a slice thickness of 0.5 mm and 29 slices with a scan time of 22 min 47 s to image the whole brain. Within the 29 scanned images, 8 images containing the hippocampal area were selected for further analysis. The hippocampal volume of each mouse was calculated using ImageJ software.

**Tissue fixation and preparation of paraffin sections**. Brain hemispheres were fixed by immersion in 4% paraformaldehyde in phosphate buffer solution (Nacalai tesque, Kyoto, Japan). Ethanol-fixed brains were embedded in paraffin, and 4 μm thick sections were mounted onto MAS-GP-coated glass slides. Sections were stained with hematoxylin and eosin (H&E) or immunostained using primary and secondary antibodies, the details of which are provided in Supplementary Table 1.

**Histochemistry**. H&E staining was carried out according to the following method. After deparaffinization, sections were stained with Mayer's hematoxylin solution (Wako, Tokyo, Japan) for 10 min, and then stained with eosin alcohol solution (Wako) for 4 min. Finally, the sections were dehydrated and coverslipped. Terminal deoxynucleotidyltransferase-mediated dUTP-biotin nick end labeling (TUNEL) staining was carried out using the DeadEnd Fluorometric TUNEL System (Promega Madison, WI), according to the manufacturer's instructions.

**Immunohistochemistry**. In experiments in which we used anti-Aβ (N1D)[60], a direct immunofluorescence method was applied. When we stained using anti-CAPON, anti-GFP, anti-phospho-tau (AT8), anti-NeuN, anti-cleaved caspase 3, anti-cytochrome C (CytC) or anti-nNOS, we applied a fluorescence-indirect tyramide signal amplification (TSA) technique (TSA System; NEN, Boston, MA). When we stained using anti-GFAP, anti-Iba1, anti-Olig-2, anti-CD31 or anti-MAP2, DAKO EnVision + System (Agilent Technologies, Santa Clara, CA) was used as the second antibody and the signal was detected using tyramide-enhanced fluorescein isothiocyanate (FITC), as for the TSA method. When we stained using anti-Synaptophysin, we used FITC-labeled antibody.

After deparaffinization, sections were heated in an autoclave at 121 °C for 5 min in 10 mM sodium citrate buffer (pH 6.0) for epitope retrieval, after which endogenous peroxidase was inactivated by 0.3% hydrogen peroxide in methanol. To block nonspecific immunoreactivity, sections were treated with the blocking solutions (TSA Biotin System kit). First antibodies, diluted in TNT buffer (0.1 M Tris-HCl, 0.15 M NaCl, 0.05% Tween20, pH 7.5), were reacted overnight at 4 °C. The sections were then washed three times in TNT buffer for 5 min, and treated with secondary antibodies. In the case of the TSA method, the sections were treated with biotinylated goat anti-mouse/rabbit IgG (1:1000 dilution, Vector Laboratories, Burlingame, CA) for 1 h, and then incubated with HRP-conjugated-avidin for 30 min (1:100 dilution in TNT, TSA System) at room temperature. Visualized was achieved with tyramide-enhanced FITC (1:50 dilution in amplification solution; supplied in the TSA System) for 10 min. When the direct immunofluorescence method was being used, the sections were treated with Alexa 488- or Alexa 546-conjugated anti-mouse/rabbit IgG (1:1000 dilution, Molecular Probes, Eugene, OR). Finally, the sections were coverslipped using ProLong Gold Antifade Mountant (Thermo Fisher Scientific), and the slides were scanned on a NanoZoomer NDP system (Hamamatsu Photonics, Iwata, Japan) with ×20 resolution, an FSX100 microscope (Olympus, Tokyo, Japan), or All-in-one Fluorescence Microscope BZ9000 (Keyence, Osaka, Japan). The signals were quantified using Metamorph Imaging Software (Molecular Devices, San Jose, CA) or Definiens Tissue Studio (Definiens, Parsippany, NJ). Fluorescence intensity of each protein is calculated as the product of average of fluorescence intensity and fluorescence area.

**Duolink**. Duolink PLA probes and PLA detection reagents were purchased from Sigma-Aldrich. Deparaffinization, epitope retrieval, inactivation of endogenous peroxidase, and blocking were performed in the same manner as that used for immunohistochemistry. Antibodies (mixture of CAPON/nNOS antibody or CAPON/Tau5 antibody), diluted in TNT buffer, were initially reacted overnight at 4 °C. The sections were then washed three times in TNT buffer for 5 min, and reacted with PLA probes (a mixture of the Anti-Mouse Plus and Anti-Rabbit Minus probes) in a humidified chamber at 37 °C. The ligation, amplification, and HRP labeling reactions were performed according to the manufacturer's instructions. Interaction signals were visualized with tyramide-enhanced FITC (1:50 dilution in amplification solution; supplied in the TSA Biotin System kit) for 10 min. The sections were then reacted with Hoechst solution, and coverslipped.

The slides were scanned on a NanoZoomer NDP system (Hamamatsu Photonics, Hamamatsu, Japan) with ×40 resolution, and signals were quantified using Definiens Tissue Studio.

**Isolation of sarkosyl-insoluble fraction**. Dissected brains were immediately frozen by liquid nitrogen, and stored at −80 °C. The sarkosyl-insoluble fraction was isolated according to the method described in Sahara et al[61]. Briefly, the hippocampus was homogenized in 10× volumes of Hsiao TBS (50 mM Tris-HCl pH 8.0, 274 mM NaCl, and 5 mM KCl) containing protease inhibitor cocktail and phosphatase inhibitor cocktail. The homogenates were centrifuged at $26,300 \times g$ for 20 min at 4 °C. The resulting supernatants were stored at −80 °C as the S1 fraction. The pellets (P1 fraction) were re-homogenized in 5× volumes of high salt and sucrose buffer (10 mM Tris-HCl, pH 7.4, 0.8 M NaCl, 10% sucrose, 1 mM EGTA) containing protease inhibitor cocktail and phosphatase inhibitor cocktail, and centrifuged at $26,300 \times g$ for 20 min at 4 °C. The resulting supernatants (S2 fraction) were incubated with 1% sarkosyl (Sigma) for 1 h at 37 °C, and then, centrifuged at $150,000 \times g$ at 1 h at 4 °C. The pellets (P3 fraction) were suspended in TE buffer (10 mM Tris-HCl, pH 8.0, 1 mM EDTA), and used as the sarkosyl-insoluble fraction.

**Western blotting**. The S1 fraction or P3 fraction (sarkosyl-insoluble fraction) of the hippocampus was used for analysis. Protein concentrations were determined using a BCA protein assay kit (Pierce, Rockford, IL). An equivalent amount of protein from each animal was mixed with a 4 x sample buffer with or without 2-mercaptoethanol, then separated by SDS-polyacrylamide gel electrophoresis, and transferred electrophoretically to a PVDF membrane (Merck Millipore, Burlington, MA). The membrane was treated with the ECL prime blocking solutions (GE Healthcare, Little Chalfont, UK), and reacted with each antibody diluted in Tris-buffered saline containing 0.05% Tween20 (TNB), pH 7.5, overnight at 4 °C. The membrane was washed three times in TNB for 5 min, and treated with HRP-conjugated anti-rabbit or anti-mouse IgG (GE Healthcare) for 1 h. Immunoreactive bands on the membrane were visualized with ECL select (GE Healthcare) and scanned with a LAS-3000mini LuminoImage analyzer (Fuji Film, Tokyo, Japan).

**RNA extraction and semi-quantitative RT-PCR**. Total RNA was extracted from brain samples using RNAiso Plus (Takara, Shiga, Japan) according to the manufacturer's instructions. We performed semi-quantitative RT-PCR for detection of both 3R- and 4R-tau mRNA as previously described[19], with minor modifications. The primer pairs used for tau were: 5′-AAGTCGCCGTCTTCCGCCAAG-3′ and 5′-GTCCAGGGACCCAATCTTCGA-3′, and for glyceraldehyde-3-phosphate dehydrogenase (GAPDH): 5′- CCATGGCACCGTCAAGGCTGA-3′ and 5′-GCCAGTAGAGGCAGGGATGAT-3′. PCR products of 288 bp and 381 bp correspond to 3R-tau and 4R-tau, respectively. RT-PCR products were detected by 24, 26, 28, 30, 32, and 34 PCR cycles to mortgage the linearity of the products, and the result was decided by 26 cycles. Band intensity was calibrated using Image J software.

**Separation of cortical neurons and cell-based assays**. Cortices and hippocampi were separated from E16–18 embryos of WT mice and placed in Neurobasal medium (Thermo Fisher Scientific). The tissues were chopped with scalpels and treated with 5 mL of 0.25% trypsin at 37 °C for 15 min with rotation. Then, 0.125 mL of 1% DNaseI was added and the solution was mixed by pipetting. After centrifugation of the tissues at 1500 rpm for 3 min, 5 mL of Hank's buffered salt solution containing 0.125 mL of 1% DNaseI was added to the pellet, and incubated at 37 °C for 5 min, with gentle movement, in a water bath. Following this, the tissues were again centrifuged at 1500 rpm for 3 min, and the resulting pellet were suspended in 15 mL of Neurobasal medium containing B27 plus supplement (Thermo Fisher Scientific) and 25 μM glutamate. The cells were filtered using a Falcon 2360 Cell Strainer with 100 μm Nylon, and seeded in cell culture plates with Neurobasal medium containing B27 and glutamate. Before use, the culture medium was first changed to Neurobasal medium without glutamate, and after 24 h, was changed to Neurobasal medium containing AAV vector.

**Mitochondrial membrane potential assay**. The assay was performed using the JC-1 Mitochondrial Membrane Potential Assay Kit (Cayman Chemical Company, Ann Arbor, MI). After 14 days in vitro, cortical neurons were treated with AAV, and 48 h later, the assay was performed according to the manufacturer's instructions. $5 \times 10^4$ cells per 1 well of a 96-well plate were used as one sample. The fluorescence intensity at emission 550 nm /excitation 590 nm was detected from healthy cells, and the intensity at emission 485 nm / excitation 535 nm was detected from apoptotic (unhealthy) cells using a Infinite Pro200 plate reader (Tecan, Männedorf, Switzerland). The ratio of the fluorescence intensity between healthy and unhealthy cells was calculated.

**Oxygen consumption rate assay**. The assay was performed using the Oxygen Consumption Rate Assay Kit based on the manufacture's instruction (Cayman Chemical Company, Ann Arbor, MI). After 15 days in vitro, cortical neurons were treated with AAV, and 48 h later the assay was performed according to the

manufacturer's instructions. $5 \times 10^4$ cells per 1 well of a 96-well plate were used as one sample. The fluorescence intensity at emission 380 nm /excitation 650 nm was recorded at 37 °C every 2 min for 140 min using an Infinite Pro200 plate reader (Tecan). Change of fluorescence intensity per 1 min while the fluorescence rise linearly was calculated as relative value of oxygen consumption rate.

**Statistical analyses.** All analyses were completed with Graphpad Prism7 Software (San Diego, CA, USA). Differences between groups were examined for statistical significance with Student's *t* test.

**Reporting summary.** Further information on research design is available in the Nature Research Reporting Summary linked to this article.

## Data availability

All relevant data are available from corresponding authors upon reasonable request. The source data underlying Figs. 1a–c, 2a–c, 3a–c, 4b, 4d, 4g–i, 5b, 6a–e, 7b–d, 7g, 8a, 8b, 8e, 8f, 9a, 9c, 9e, 10a, 10b, 10d, Supplementary figure 2b, 2d, 2e, 4, 8, 9, and 10 were provided in Supplementary information files (Supplementary figures 14–32) and Source data file. Tau-interacting proteins identified by tau interactome were summarized in Supplementary Data 1. Antibodies used for western blot (WB) and immunohistochemical (IHC) analyses were shown in Supplementary Table 1.

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

## Acknowledgements
We thank Hiroki Sasaguri (RIKEN Center for Brain Science) for beneficial discussion; Yukiko Nagai (RIKEN Center for Brain Science) for secretarial work; Ryo Fujioka, and Research Resource Division (RIKEN Center for Brain Science) for technical assistance; and members of the Laboratory for Proteolytic Neuroscience (RIKEN Center for Brain Science) for beneficial suggestions. The authors also thank Mika Ito and Naomi Takino (Jichi Medical University) for their help with the production of the AAV vectors. We are grateful to Dr. Akihiko Takashima (Gakushuin University) for providing the Wtau-Tg mouse, to Virginia M.-Y. Lee (University of Pennsylvania) for providing P301S-Tau-Tg mice, to Dr. Norihiro Kato and Tadashi Okamura (National Center for Global Health and Medicine) for providing CAPON-deficient mice, and to Peter Davis (Litwin Zucker Center for Alzheimer's Research, Long Island) for providing PHF-1 and MC1 antibody. A part of this work was achieved by collaboration with Astellas Pharma Inc, and we thank Yoshitsugu Shitaka, Mitsuyuki Matsumoto, Yasuyuki Mitani, Hiroshi Yamada, Shinich Miyake, Mayuko Okabe, Mayako Yamazaki, and Ni Keni for helpful suggestions.

This work was supported by a Grant-in-Aid for Young Scientists (B) (a Ministry of Education, Culture, Sports, Science and Technology (MEXT) grant) (15K19036) (SH), and research grants from the RIKEN Special Postdoctoral Research program (SH). Support was also received via a Grant-in-Aid for Scientific Research (B) (a MEXT grant) (TS), and by the Brain Mapping by Integrated Neurotechnologies for Disease Studies (Brain/MINDS) from the Japan Agency for Medical Research and Development (AMED) (TCS) (JP18dm0207001), as well as from a research grant from the RIKEN Center for Brain Science.

## Author contributions
S.H., Y.M., N.K., N.M., N.S., and J.T. performed experiments. S.M. produced and provided AAV vectors for the project. S.H., T.S. and T.C.S. supervised the research. S.H. and T.C.S. wrote and edited the manuscript.

## Additional information

**Competing interests:** S.H., Y.M., T.S. and T.C.S. serve as a member, a director, an advisor and a CEO, respectively, for RIKEN BIO Co. Ltd. S.M. servers as a director and owns equity in a gene therapy company (Gene Therapy Research Institution Co., Ltd.). The remaining authors declare no competing interests.

