## [Peer Review File · Nature Communications]

Reviewers' comments:

Reviewer #1 (Remarks to the Author):

Overview:

This study by Hashimoto et al. describes that A β pathology leads to the accumulation of CAPON protein, and that increase of CAPON induces tau pathology and neuronal cell death, suggesting that CAPON might be the missing link that's connecting A β and Tau pathology. However, it is not clear from their results about which cells CAPON may play a major role during AD and related dementia (ADRD). It will be also interesting to see what kind of cell death (i.e. pyroptosis and necrosis) is happening during ADRD apart from Caspase-3 mediated apoptotic cell death, considering CAPON's link to neuroinflammation and a recent study showing caspase-3 cleavage of a gasdermin can induce pyroptosis and necrosis (Nature. 2017 Jul 6;547(7661):99-103.).

Major points:

1. CAPON is upregulated in AppNL-G-F/NL-G-F mice and downregulated in P301S mice. How about in the double knock-in mice (AppNL-G-F/NL-G-F mice crossbred with MAPT-KI mice) or any other tau transgenic mice?
2. Fig 2: CAPON expression patterns upon LPS injection vary by fields, with some possibly showing vascular structures and also CAPON doesn't show perfect overlap with nNOS either. In fact, nNOS is not only expressed in neurons, but also astrocytes, oligodendrocytes, and endothelial cells of the blood vessels. Therefore, cell specific-expression of CAPON in these AD mouse models needs to be more carefully examined using markers for neurons, astrocytes, oligodendrocytes, and endothelial cells. Without the labeling with NeuN marker for neurons, it's hard to say "CAPON was increased in neuronal rather than glial cells (Fig2c, right)." Also, quantifications of the co-localization of Iba1/CAPON and nNOS/CAPON are missing.
3. Since CAPON is a tau-interacting protein, what about the cell-specific expression of CAPON and its interaction with nNOS in tau-transgenic mice?
4. Synapsin I promoter will mainly express CAPON in neuronal cells. Considering potential expression of CAPON in other cell types, using other promoters will be equally important to study the role of CAPON.
5. In Fig.4d, there is a significant number of Caspase-3 positive cells that are not NeuN positive. What is the cell identity of these cells?
6. In Fig. 4e, what kind of cells are these TUNEL-positive cells? Also considering the massive amount of Caspase-3-positive cells in Fig.4d, there aren't many TUNEL-positive cells shown here, suggesting not all the Caspase-3-positive cells are going through apoptosis.
7. Quantifications are missing for Fig. 4f and g.
8. How does CAPON overexpression increase tau pathology? Is it through hyperphosphorylation by ERK1/2? Can ERK1/2 inhibitors reduce the increased tau pathology upon CAPON overexpression?
9. In addition to overexpressing CAPON, it's also important to study the effect of knock out CAPON in a mouse model of AD (where CAPON expression is upregulated) or at least in cell culture model to see if CAPON deficiency can rescue the AD pathology.
10. Does neuroinflammation change upon CAPON overexpression, considering neuroinflammation also contributes to tau pathology?
11. Fig. S5 and elsewhere - a clear rationale for using different age group (9 vs 15 months for example in Fig. S5) is needed.
12. If the double-KI (APP or tau) doesn't differ from WT in showing CAPON induced cell death, then, why were the mitochondrial damage, alterations in pERK/BAX/pMEK and Destras 1 was assessed in double-KI mice (which may be confounding as Abeta and/or tau could potentially alter these markers directly – independent of CAPON).

Minor points:

1. In the Results section, nNOS involvement and the rational to check nNOS isn't discussed in detail with references.

2. In Figure 2C, the merged images of CAPON/Iba1 are shifted, making it difficult to compare with the single channel images.
3. Introduction (last paragraph): Shouldn't this be APP NL-F and APP NL-G-F?
4. Question on IP/LCMS: Not quite clear why IP was done from; 1) the whole brain, instead of hippocampus; 2) rationale for choosing 9-month-old mice.
5. Fig. 1a - is it NL-GF or NL-G-F? Blot labeling is different from bar labeling in the histogram.
6. Results referring to Fig. 1a/1b: (the sentence "CAPON expression pattern enhances the progression of AD pathology...") - does not really show CAPON expression enhances the progression of AD pathology. This needs to be edited.
7. Fig. 5C - is not mitochondrial membrane potential (as described in the results). It should be Fig. 5B. Similarly, Dextras1 data is in 5C (not 5B).
8. Fig.6a: Is the CAPON-Tau interaction (Duolink) also detectable in WT mice? This is important to understand if it is something specific to human tau vs mouse tau. Moreover, double-KI mice don't show any cell death anyway.
9. Fig. 8a. Tau blot is necessary to confirm tau KO.
10. Discussion: "A recent study used APP/PS1..." sentence needs the reference.
11. Fig. S2C - last panel, genotype information on the top right corner is missing symbols ('-/-').
12. Fig5-Legend: is it 'mitochondrial membrane potency' or 'mitochondrial membrane potential'?

Reviewer #2 (Remarks to the Author):

The present manuscript focuses on the role of a novel Tau-binding protein, CAPON, into the pathophysiological development of Alzheimer's Disease using newly developed knock-in models. Overall, although the topic is of interest, to my eyes the manuscript is not structured enough and systematic to have a clear view of what is going on.

General comments:

First, it is not clear why authors use double KI mice if CAPON impacts on hippocampal structure and pTau/Tau aggregation in WT animals. Actually, to draw clear conclusion about the physiopathological impact of CAPON upregulation, overexpression should be performed in WT, APP KI, Tau KI and double KI animals with a systematic evaluation of Abeta accumulation and/or Tau pathology using reliable quantitative histological and biochemical methods. Also, authors mentioned a publication under consideration related to APP/Tau double KI mice but this paper is not provided making difficult to appreciate the interest of this model vs. individual KI. Age of animals is not always indicated. For instance, it is not clear when AAV CAPON overexpression takes place vs the pathophysiological development of models and vs. CAPON overexpression in APP model.

Second, a lot of data are not quantified (particularly immunohistochemical data) and number of replicates is often fair.

Third, no behavioral readout is provided making difficult to conclude about the impact of CAPON towards Abeta and Tau-induced behavioral changes providing that memory alterations is a final readout not always associated with lesional changes.

Specifically:

1) Interaction of Tau with CAPON has been uncovered from IP of FLAG from 2N4R human WT Tau overexpressed under the control of a CAMKII promoter. Author should prove the interaction by Co-IP and PLA in their new Tau KI model in order to demonstrate the reality of the interaction in a more physiological context. Indeed, it cannot rule out that the FLAG located at the Cter Tau part

impacts on IP experiments.

2) It is not clear and discussed why CAPON levels decrease in P301S mice. This is an aggressive model and given CAPON is likely a neuronal protein, it could be that CAPON reduction parallels neurodegeneration in this model. What about CAPON expression in Tau KI mice ?

3) Figure 2c. No quantification is provided and, even there are obvious differences between left panels, no clear indication is provided. Stating that based on this LPS experiments in WT mice that CAPON upregulation is due to neuroinflammation in APP KI mice is a bit overstated. To answer this question, one may block neuroinflammation in APP KI mice and check CAPON.

4) AAV-CAPON overexpression has been performed in APP - Tau KI mice and APP KI mice but what about Tau KI mice alone. The approach should be systematic in all genotypes to draw a clear conclusion. Also, it is not clear when AAV is injected. No quantification for NeuN, Caspase 3, TUNEL, GFAP, Iba1 is provided. No comparison on the effect of CAPON on these parameters in all genotypic group which help to draw a conclusion on synergies with Abeta and Tau.

5) CAPON overexpression is toxic by itself. Authors suggest it may be related to mitochondrial changes. However, this is not well exemplified. First, in vivo, cytochrome c delocalization should be demonstrated using cytosolic fractionations. In vitro, authors should used more function mitochondrial relevant measurement, for instance using Seahorse. At this stage, mechanisms are not clearly defined. If apoptosis is relevant, this should be proven also in vivo.

6) In Figure 6, it is not clear if Total Tau protein is altered or not. Blots show this is the case while quantifications do not. Oligomers involvement should be proven by immunohistochemistry. Conformational Tau should be addressed by immunihistochemistry (MC1, AT100....). Immuno data should be quantified.

Responses to the reviewers' comments*

The title: A novel tau binding protein: CAPON induces neurodegeneration in an Alzheimer's disease mouse model (NCOMMS-19-00318-T)

*Additions and changes made in the revised manuscript are indicated by yellow.

Answer to Reviewer #1:

We thank the reviewer for the valuable comments. We have revised our paper accordingly and feel that reviewer's comments helped clarify and improve our paper. Please find our responses (indicated by blue letters) to reviewer's specific comments (indicated by black letters) below.

Overview:

This study by Hashimoto et al. describes that A β pathology leads to the accumulation of CAPON protein, and that increase of CAPON induces tau pathology and neuronal cell death, suggesting that CAPON might be the missing link that's connecting A β and Tau pathology. However, it is not clear from their results about which cells CAPON may play a major role during AD and related dementia (ADRD). It will be also interesting to see what kind of cell death (i.e. pyroptosis and necrosis) is happening during ADRD apart from Caspase-3 mediated apoptotic cell death, considering CAPON's link to neuroinflammation and a recent study showing caspase-3 cleavage of a gasdermin can induce pyroptosis and necrosis (Nature. 2017 Jul 6;547(7661):99-103.).

We appreciate the extremely helpful suggestion regarding the cell death mechanisms caused by the CAPON overexpression. The mechanisms of cell death appear quite complicated as follows. We observed significant activation of Gasdermin D (GSDMD) and Gasdermin E (GSDME) in mice treated with AAV-CAPON. GSDMD acts as an essential effector of pyroptosis, which represents an inflammatory form of cell death. GSDME, activated by caspase 3 in apoptotic pathway, induces necrosis and pyroptosis secondarily when apoptotic cells are not scavenged (Wang et al, Nature, 2017). Therefore, CAPON overexpression-mediated cell death cannot be accounted for by one simple pathway but rather by more complicated mechanisms, which involve pathways of both apoptosis and pyroptosis. We have added the data to Fig. 4f and discussed the results in page 6 (Result part) and page10 (Discussion part).

Major points:

1. *CAPON is upregulated in AppNL-G-F/NL-G-F mice and downregulated in P301S mice. How about in the double knock-in mice (AppNL-G-F/NL-G-F mice crossbred with MAPT-KI mice) or any other tau transgenic mice?*

We compared the expression levels of CAPON in WT, and MAPT (hTau)-KI, double-KI mice. As a result, MAPT (hTau)-KI showed almost identical CAPON levels as compared to WT mice, whereas double-KI showed significantly higher CAPON levels than single MAPT (hTau)-KI mice. These results indicate that humanization of tau itself does not affect CAPON levels and that amyloid pathology increases CAPON levels also in WT and MAPT (hTau) KI mice. These results have been added in the new Fig. S3. We consider that neurodegeneration observed in P301S-Tau-Tg mice decreases CAPON levels because CAPON is mainly expressed in neuronal cells. CAPON levels appear decreased after the mice undergo neurodegeneration. In contrast, the CAPON levels appear unchanged in such mice as MAPT (hTau) KI, mice which do not display neurodegeneration.

2. *Fig 2: CAPON expression patterns upon LPS injection vary by fields, with some possibly showing vascular structures and also CAPON doesn't show perfect overlap with nNOS either. In fact, nNOS is not only expressed in neurons, but also astrocytes, oligodendrocytes, and endothelial cells of the blood vessels. Therefore, cell specific-expression of CAPON in these AD mouse models needs to be more carefully examined using markers for neurons, astrocytes, oligodendrocytes, and endothelial cells. Without the labeling with NeuN marker for neurons, it's hard to say "CAPON was increased in neuronal rather than glial cells (Fig2c, right)." Also, quantifications of the co-localization of Iba1/CAPON and nNOS/CAPON are missing.*

We have carefully analyzed localization of "endogenous", "LPS-introduced" and "AAV-treated" CAPON by co-immunohistostaining with several cell-specific markers (Figs. S5 and S6). As a result, none of the three types of CAPON: "endogenous", "LPS-introduced" and "AAV-overexpressed" colocalized with Iba1 (microglia), GFAP (astrocyte), Olig2 (Oligodendrocyte), and CD31 (endothelial cells) signals. In contrast, we observed partial colocalization of CAPON expression with NeuN (neuronal cell bodies), VGAT (pre-synapse) or Synaptophysin (pre-synapse). These results indicate that endogenous CAPON mainly localizes in neuronal cells and that LPS increases CAPON level in neuronal cells, but not in other type of cell types. Consistently, SR. Jaffrey et al (Neuron, 1999) also observed focal expression of endogenous CAPON in neuronal cells. We do agree however with the reviewer that we had overstated in the previous manuscript. We have now toned down as follows: "CAPON was

increased in neuronal rather than glial cells (Fig2c, right).” to “CAPON appears partially increased in neuronal rather than glial cells (Fig2c, right)” (page5). Also, we have added the quantification values of overlapping CAPON/Iba1 and CAPON/nNOS in Fig. 2c.

3. Since CAPON is a tau-interacting protein, what about the cell-specific expression of CAPON and its interaction with nNOS in tau-transgenic mice?

As we mentioned in the preceding response, endogenous CAPON is mainly expressed in neuronal cells (Fig. S5). Besides, expression pattern of CAPON is almost identical between WT and tau-transgenic mice (see the Appendix 1 at the end of this document). Therefore, we speculate that the CAPON-nNOS interaction takes place in neuronal cells both in WT and tau-transgenic mice. It is consistent that SR. Jaffrey et al (Neuron, 1999) demonstrated that CAPON interacts with PSD95, implying the presence of CAPON in the synaptic dendrites. This supports the idea that CAPON interacts with nNOS in neuronal cells.

4. Synapsin I promoter will mainly express CAPON in neuronal cells. Considering potential expression of CAPON in other cell types, using other promoters will be equally important to study the role of CAPON.

We agree with reviewer that various promoters can be tested. It would be very interesting to examine the effect of CAPON overexpression in other type of cells. However, we did not perform overexpression of CAPON using other promoters than Synapsin I promoter for the following reasons:

1. In this study, we focused on CAPON as a tau-interacting protein. Since tau pathology arises in neuronal cells in AD, we considered that CAPON expressed in neuronal cells plays an important role in tau-related neuronal cell death in AD.
2. As we mentioned above, expression of “endogenous” and “LPS-induced” CAPON was mainly detected in neuronal cells. In addition, Hashimoto et al (J. Cell. Mol. Med., 2012) previously indicated that CAPON is increased in hippocampal pyramidal cells in AD brain. Therefore, CAPON expressed in neuronal cells is of primary importance for pathogenesis of AD.

There are a number of potential promoters that can be tested. Promoters for non-neuronal cells and for interneurons such as cholinergic neurons are very interesting, but there are too many to test, and it will take years to do so. It is thus beyond the scope of the present study. We will make such investigations in the future. We thank the reviewer for the important suggestion.

5. In Fig.4d, there is a significant number of Caspase-3 positive cells that are not NeuN positive. What is the cell identity of these cells?

Because NeuN signal is detected only in soma of neuronal cells (not in axons and dendrites), it is hard to indicate that the other signals are not present in neurons even if the signals do not colocalize with NeuN. We have carefully analyzed localization of cleaved-caspase 3 in CAPON-overexpressing mice brain by immunohistochemistry with other neuronal cell markers (Fig. S7). We observed partial colocalization of cleaved caspase-3 not only with NeuN but also with MAP2 (dendrites), suggesting that caspase 3-mediated cell death took place in neuronal cells in mouse brain treated with CAPON-AAV. In addition, since the NeuN signal, but not Iba1 and GFAP signals, is reduced in CAPON-overexpressing mice (Figs. 4d, g and h), it is most likely that the cell death took place in neuronal cells, but not in glial cells.

6. In Fig. 4e, what kind of cells are these TUNEL-positive cells? Also considering the massive amount of Caspase-3-positive cells in Fig.4d, there aren't many TUNEL-positive cells shown here, suggesting not all the Caspase-3-positive cells are going through apoptosis.

Co-staining of TUNEL and cell-specific-markers should provide an answer, but it is difficult for technical reasons as the reviewer would agree. However, we speculate that TUNEL positive cells are neuronal cells because cleaved-caspase 3 positive cells are neuronal cells and NeuN signal is decreased in CAPON-overexpressing cells. As the reviewer pointed out, the number of TUNEL-positive cells is smaller than that of cleaved-caspase 3 positive cells. This is presumably because the cell death mechanisms induced by CAPON overexpression is not just a simple classical apoptotic pathway and comprises of complex mechanisms which involve multiple pathways without accompanying major DNA fragmentation. It may be also important that TUNEL is sometimes not very sensitive particularly when fragmented DNA is further degraded and cleared up. This might explain why there are not so many TUNEL-positive cells in our case as well as in general.

7. Quantifications are missing for Fig. 4f and g.

We have quantified the signals and the values are shown in graphs.

8. How does CAPON overexpression increase tau pathology? Is it through hyperphosphorylation by ERK1/2? Can ERK1/2 inhibitors reduce the increased tau pathology upon CAPON overexpression?

We do not believe that ERK induces tau phosphorylation in CAPON-overexpressing mice, but rather it appears that activation of the ERK signaling links to mitochondrial dysfunction in CAPON-overexpressing mice. As we mentioned in discussion part (page 11), we consider that facilitation of NO-dependent nitration or neuroinflammation mediate CAPON increased tau pathology.

9. In addition to overexpressing CAPON, it's also important to study the effect of knock out CAPON in a mouse model of AD (where CAPON expression is upregulated) or at least in cell culture model to see if CAPON deficiency can rescue the AD pathology.

We thank the reviewer for the extremely important suggestion. We agree that it is important to investigate the effect of CAPON deficiency on the AD-related pathological phenotype. In the additional experiments, we crossbred CAPON KO mice with P301S-Tau-Tg tauopathy model mice and analyzed their AD-related pathology. As a result, we observed significant attenuation of tau phosphorylation, tau aggregation, and brain atrophy in 12-month-old CAPON^{-/-} X P301S-Tau-Tg mice as compared to P301S-Tau-Tg mice. These results consistently support the evidence that CAPON is closely associated with tau pathology and neuronal cell death and suggest that CAPON induces neuronal cell death not only via tau-independent mechanism but also via tau-dependent mechanism. The results have been added to Fig. 9.

10. Does neuroinflammation change upon CAPON overexpression, considering neuroinflammation also contributes to tau pathology?

It is not very easy to answer this question in a definite manner because “neuroinflammation” itself is not perfectly defined and because many aspects remain unknown, but previous studies indeed demonstrated contribution of neuroinflammation to tau pathology. For instance, mutations in TREM2, an immunoglobulin receptor expressed by microglia, increases the risk of AD and is accumulated associated with tau pathology and synaptic loss in post mortem human cortical samples from AD patients (Laurent et al, Biochem.J.,2017). Also, according to Asai et al (Asai et al, Nat. Neurosci., 2014), depletion of microglia and inhibition of exosome synthesis interrupted tau propagation, suggesting that microglia and exosomes contribute to the progression of tauopathy. Although there is still room for argument about relationship between neuroinflammation and tau pathology, there appears to be a strong linkage. We thus speculated that CAPON overexpression may also facilitate tau pathology via neuroinflammation. We have added the discussion in page 11.

11. *Fig. S5 and elsewhere - a clear rationale for using different age group (9 vs 15 months for example in Fig. S5) is needed.*

In the hippocampi of *App^{NL-G-F}* (and double-KI) mouse, amyloid plaque formation starts from around 2-month-old, and plateau at around 12-month-old. We carried out experiments using both 9- and 15-month-old mice in order to adjust effect of amyloid formation on CAPON action. (9-month-old mice are at a developmental stage in terms of amyloid pathology, 15-month-old are at a saturated stage.) Consequently, we detected no difference in the action of CAPON overexpression on their brains (tau-phosphorylation, neuronal cell death, and hippocampal atrophy) between 9 and 15-month-old double-KI mice.

12. *If the double-KI (APP or tau) doesn't differ from WT in showing CAPON induced cell death, then, why were the mitochondrial damage, alterations in pERK/BAX/pMEK and Dexas 1 was assessed in double-KI mice (which may be confounding as Abeta and/or tau could potentially alter these markers directly – independent of CAPON).*

Because we observed no significant induction of neuronal cell death in *App*-KI mice (Saito et al, Nat. Neurosci., 2014) or hTau-KI mice (Saito et al, under consideration) compared to WT mice, we assumed that the *App*-KI and hTau-KI mice do not display drastic mitochondrial damage and MEK/ERK/Dexas1 alteration. However, we believe that we should determine the activation status of MEK pathway in double-KI, WT, and CAPON-overexpressing WT mice as controls for each other. MEK/ERK and Dexas1 remained unaltered in the double KI mice as compared to WT (Fig. S11a). In addition, we also observed MEK activation and Dexas1 reduction in CAPON-overexpressing WT mice (Fig. S11b) as well as double-KI mice. These results suggest that the MEK/ERK/Dexas1 pathway is not significantly activated at least by amyloid pathology or by tau humanization.

Minor points:

1. *In the Results section, nNOS involvement and the rational to check nNOS isn't discussed in detail with references.*

We now discuss the point with references (page 5).

2. *In Figure 2C, the merged images of CAPON/Iba1 are shifted, making it difficult to compare with the single channel images.*

We thank the reviewer for the note. We have now corrected the figure.

3. Introduction (last paragraph): Shouldn't this be APP NL-F and APP NL-G-F?

We mean “App^{NL-G-F} and App^{NL-G-F} X MAPT double-KI”, but not “App^{NL-F} and App^{NL-G-F}” in last paragraph of introduction (page 4) because we do not use NL-F mice in the present study.

4. Question on IP/LCMS: Not quite clear why IP was done from; 1) the whole brain, instead of hippocampus; 2) rationale for choosing 9-month-old mice.

- 1. Answer to Q1: We used whole the brain after removal of the olfactory bulb and cerebellum for IP/MS because the quantity of protein from hippocampus was too small for IP and comprehensive analysis. We also wanted to minimize the time for dissection in order to avoid protein degradation and non-specific protein-protein interaction. Since tau pathology is seen not only hippocampus and cortical region but also amygdala and thalamic region in advanced AD brain, we believe that use the whole brain without olfactory bulb and cerebellum for the experiment is reasonable. We have specified the details of brain samples for IP/MS in Method section (page 14).**
- 2. Answer to Q2: As reported in the original paper describing WT tau-Tg mice (Kimura et al, EMBO J.,2007), we detected hyperphosphorylated tau approximately from 8 months of age, and this increased with aging of the Tg mice. The original paper also showed that the aged mice (20-24 month-old), but not adult mice (9-13 month-old), exhibited a significant decline of the neuronal activity detected by Mn-enhanced MRI and impairment of memory tested by the Morris Water Maze. Because our intension has been to identify tau-interacting proteins, which aggravate tau pathology and accelerate neuronal cell death in the early stage before cognitive impairment, we choose 9-month old Tg mice which are in the pre-memory impairment stage.**

5. Fig. 1a - is it NL-GF or NL-G-F? Blot labeling is different from bar labeling in the histogram.

We have unified the labeling to “NL-G-F”.

6. Results referring to Fig. 1a/1b: (the sentence “CAPON expression pattern enhances the progression of AD pathology...” - does not really show CAPON expression enhances the progression of AD pathology. This needs to be edited.

We agree and thus have eliminated the sentence.

7. Fig. 5C - is not mitochondrial membrane potential (as described in the results). It should be Fig. 5B. Similarly, Dextras1 data is in 5C (not 5B).

We thank the reviewer for the indication. We have corrected the figure numbers.

8. Fig.6a: Is the CAPON-Tau interaction (Duolink) also detectable in WT mice? This is important to understand if it is something specific to human tau vs mouse tau. Moreover, double-KI mice don't show any cell death anyway.

We also detected the CAPON-tau interaction in all of WT and hTau single KI mice by Duolink, suggesting that both murine and human tau can interact with endogenous CAPON. The data has been added to Fig. S12.

9. Fig. 8a. Tau blot is necessary to confirm tau KO.

Western blot data of tau immunoreactivity for WT and Tau KO mouse brains is now shown in Fig. S25.

10. Discussion: "A recent study used APP/PS1..." sentence needs the reference.

We have added the reference.

11. Fig. S2C - last panel, genotype information on the top right corner is missing symbols ('-/-').

We have accordingly corrected the figure.

12. Fig5-Legend: is it 'mitochondrial membrane potency' or 'mitochondrial membrane potential'?

We thank the reviewer for making the point. We have corrected the wording to "membrane potential".

Answer to Reviewer #2:

We appreciate the valuable comments. We have revised our paper accordingly and feel that reviewer's comments helped improve our paper. Please find our responses to reviewer's specific comments below.

The present manuscript focuses on the role of a novel Tau-binding protein, CAPON, into the pathophysiological development of Alzheimer's Disease using newly developed knock-in models. Overall, although the topic is of interest, to my eyes the manuscript is not structured enough and systematic to have a clear view of what is going on.

General comments:

First, it is not clear why authors use double KI mice if CAPON impacts on hippocampal structure and pTau/Tau aggregation in WT animals. Actually, to draw clear conclusion about the physiopathological impact of CAPON upregulation, overexpression should be performed in WT, APP KI, Tau KI and double KI animals with a systematic evaluation of Abeta accumulation and/or Tau pathology using reliable quantitative histological and biochemical methods. Also, authors mentioned a publication under consideration related to APP/Tau double KI mice but this paper is not provided making difficult to appreciate the interest of this model vs. individual KI. Age of animals is not always indicated. For instance, it is not clear when AAV CAPON overexpression takes place vs the pathophysiological development of models and vs. CAPON overexpression in APP model.

First, we chose the double KI mice for the overexpression experiments for the following reasons.

- 1. We observed an increase in CAPON levels under amyloid pathology (Fig. 1). According to the amyloid cascade hypothesis, amyloid pathology is followed by tau pathology and neurodegeneration in AD. Therefore, to investigate the effect of CAPON accumulation on tau or neurodegeneration, we should use amyloid-based mouse.**
- 2. hTau-KI brains, which expresses all six isoforms of humanized tau, are more similar to humans because adult mouse brains express only three isoforms. Also, humanization of tau itself does not alter tau pathology, neuronal cell death (Saito et al, under consideration) and CAPON levels (shown in Fig. S3). After all, there was no difference between WT and hTau knock-in mice in terms of the effect of CAPON overexpression, this was unpredictable when the research plan was started. Thus, we used double-KI to investigate CAPON function in more physiologically relevant conditions. However, as reviewer pointed out, we should performed basic investigations using hTau-single-KI**

mouse. The details of the results of the investigations are described in comment #4 in the “Specifically” part.

Second, we have specified age, sex of animals, and period of AAV-introduction in each of the figure legends.

Second, a lot of data are not quantified (particularly immunohistochemical data) and number of replicates is often fair.

We have added quantified values to each immunohistochemical data in main figures.

Third, no behavioral readout is provided making difficult to conclude about the impact of CAPON towards Abeta and Tau-induced behavioral changes providing that memory alterations is a final readout not always associated with lesional changes.

Since AAV injection-mediated gene introduction leads to quite variable overexpression levels of the target protein, to perform behavioral tests is not practical. (We would need an extremely large number of animals.) As the reviewer pointed out, memory impairment may not always be associated with lesional changes, but neurodegeneration in humans is clearly a direct cause of clinical symptom in AD. In the study, we aim to clarify the role of CAPON in pathological alteration and to provide a mechanistic insight. Therefore, we believe that the behavioral experiments, which usually involve multi-factorial parameters, are beyond the scope of present study and that functional analysis of neurocircuits may be a more precise strategy.

Specifically:

1) Interaction of Tau with CAPON has been uncovered from IP of FLAG from 2N4R human WT Tau overexpressed under the control of a CAMKII promoter. Author should prove the interaction by Co-IP and PLA in their new Tau KI model in order to demonstrate the reality of the interaction in a more physiological context. Indeed, it cannot rule out that the FLAG located at the Cter Tau part impacts on IP experiments.

To our knowledge, none of the anti-tau antibodies available are suitable for immunoprecipitation with adequate avidity and specificity (Appendix2). We thus utilized Duolink (PLA) system and successfully detected the signals of CAPON-tau interaction in WT and MAPT (hTau)-single-KI mouse where tau exists without the FLAG tag. The Duolink data sets have been added to Fig. 2a and Fig. S12a.

2) *It is not clear and discussed why CAPON levels decrease in P301S mice. This is an aggressive model and given CAPON is likely a neuronal protein, it could be that CAPON reduction parallels neurodegeneration in this model. What about CAPON expression in Tau KI mice?*

As the reviewer pointed out, total amount of CAPON protein could be decreased by progression of neuronal cell death in P301S-Tau-Tg mouse and AD brain since CAPON is mainly expressed in neuronal cells (Fig. S5). As CAPON is increased under amyloid pathology, CAPON appears increased in early stage of AD and then reduced with the advance of neurodegeneration in later stage. Since single hTau-KI shows no apparent AD pathology (A β , Tau and neuroinflammation) and neuronal cell death, we speculated CAPON levels are unaltered in hTau-KI mice as compared to WT mice. Indeed, we observed no difference of CAPON expression levels between WT and hTau-KI mice. The data has been added to Fig. S3.

3) *Figure 2c. No quantification is provided and, even there are obvious differences between left panels, no clear indication is provided. Stating that based on this LPS experiments in WT mice that CAPON upregulation is due to neuroinflammation in APP KI mice is a bit overstated. To answer this question, one may block neuroinflammation in APP KI mice and check CAPON.*

We quantified the signals of Fig. 2c (left) and added the values to the graph. We also showed quantified values for colocalization of CAPON/Iba1 and CAPON/nNOS (Fig. 2c, right). We agree with the reviewer that our previous conclusion was a an overstatement. We thus have toned down and changed the conclusion to “neuroinflammation associated with A β pathology may possibly be involved in CAPON elevation” (page 5).

4) *AAV-CAPON overexpression has been performed in APP - Tau KI mice and APP KI mice but what about Tau KI mice alone. The approach should be systematic in all genotypes to draw a clear conclusion. Also, it is not clear when AAV is injected. No quantification for NeuN, Caspase-3, TUNEL, GFAP, Iba1 is provided. No comparison on the effect of CAPON on these parameters in all genotypic group which help to draw a conclusion on synergies with Abeta and Tau.*

We described the reasons why we used double-KI mice in this study in the answer to “General Comments.” We do however agree with the reviewer’s opinion. Initially, we performed ICV injection of AAV to mice and analyzed pathologies 3 months after the injection. We have now performed direct hippocampal introduction of AAV, which can induce rapid overexpression of the target protein, because the time give for the revision was limited. AAV was injected into

hippocampi of hTau single KI and *App*/hTau double KI mouse brains, and we then analyzed their pathological changes 3 weeks after the injection. We observed induction of neuroinflammation, neuronal loss and activation of caspase 3 both in hTau-single-KI and double-KI. These results suggest that CAPON overexpression induces neuronal cell death in hTau-KI as well as double-KI. The data has been added to Fig. S9. In response to the reviewer's comments, we specified age, sex of animals, and period of AAV-introduction in each figure legend. We also performed quantitative analysis for immunohistochemistry, and the values have been added to each figure.

5) *CAPON overexpression is toxic by itself. Authors suggest it may be related to mitochondrial changes. However, this is not well exemplified. First, in vivo, cytochrome c delocalization should be demonstrated using cytosolic fractionations. In vitro, authors should used more function mitochondrial relevant measurement, for instance using Seahorse. At this stage, mechanisms are not clearly defined. If apoptosis is relevant, this should be proven also in vivo.*

Using an *in vitro* paradigm, we analyzed oxygen consumption rate of CAPON-overexpressing primary neurons using Oxygen Consumption Rate Assay Kit (Cayman). We observed significant attenuation of oxygen consumption in CAPON-overexpressing cells but not in GAPDH-overexpressing cells, suggesting that CAPON overexpression indeed caused a mitochondrial damage in neuronal cells. In an *in vivo* paradigm, we determined the levels of cytochrome c by Western blot analysis using the cytosolic fraction. We detected a significant overproduction of cytochrome c in CAPON mouse. The overproduction was not overt presumably because the release of cytochrome c took place locally while the entire protein extract of whole hippocampus was subjected to the biochemical analysis. The data of oxygen consumption assay and cytochrome c western blot have been shown in Figs. 5d and b, respectively.

We believe that activation of MEK/ERK signal from CAPON/Dexas1 induces mitochondrial dysfunction. In addition, CAPON-induced cell death pathways could also contribute to mitochondrial damage. As for the cell death mechanism caused by CAPON, we have shown that caspase 3 is activated by CAPON overexpression. Additionally, we also observed significant activation of Gasdermin D (GSDMD) and Gasdermin E (GSDME) in mice treated with AAV-CAPON. GSDMD acts as an essential effector of pyroptosis, an inflammatory form of cell death. GSDME is activated by caspase 3 in apoptotic pathway and induces necrosis and pyroptosis secondarily when apoptotic cells are not scavenged (Wang et al, Nature, 2017). So, CAPON overexpression-mediated cell death cannot be attributed only to a single pathway but

also to complicated mechanisms which involve apoptosis and pyroptosis (Fig. 4f). Mitochondrial dysfunction is likely activated by these complex pathways.

6) In Figure 6, it is not clear if Total Tau protein is altered or not. Blots show this is the case while quantifications do not. Oligomers involvement should be proven by immunohistochemistry. Conformational Tau should be addressed by immunohistochemistry (MC1, AT100....). Immuno data should be quantified

We did not observe extensive tau aggregates in CAPON-overexpressing mice in a manner similar to that in AD or the tauopathy model. As requested by the reviewer pointed out, we performed immunohistochemistry using MC1 antibody and observed MC1 positive cells, which are also immuno-reactive for phosphorylated-S404 antibody in hippocampal pyramidal cell layer. From these results, we assume that CAPON-overexpressing mice show an early stage of tau pathology. The results have been added to Fig. 6f. Naruhiko Sahara, who performed these experiments and analyzed the data, is now included as a new co-author.

CAPON/Hoechst

WT (12M)

P301S-Tau-Tg (12M)

Appendix 2

Protein samples: Tris-HCl soluble fraction of hippocampus and cortex from WT and Wtau-Tg mouse

IP: Tau5 (total tau) antibody

WB: Tau5 (total tau) antibody

Tau was not immunoprecipitated by tau5 antibody from WT sample.

Reviewers' comments:

Reviewer #1 (Remarks to the Author):

Summary

The authors have significantly revised the manuscript and satisfactorily answered the previously raised questions by looking up Gasdermin expressions and cell specific-expression of CAPON and also knocking out CAPON in the P301S-Tau-Tg tauopathy model.

Major points:

Since CAPON deficiency seem to selectively affect hippocampal neurons, functional analysis in terms of behavioral studies may be warranted (if possible). Especially, since this is the first report of CAPON's role in Alzheimer's disease and related tauopathies, if the authors have done or likelihood of performing any behavioral/cognitive analysis, it may be important to include to strengthen their findings.

Minor points:

1. Figure 3 can be moved to the supplementary figures since it is mainly showing the validation of AAV-overexpression model.
2. There are some imperfections in the way the blots are presented. For example, black lines surrounding the blots are missing sometimes, which makes it inconsistent.
3. Specify which co-localizations are shown by Duolink in the figures, so that it will be easier for the easy readability.

Reviewer #2 (Remarks to the Author):

--

Answer to Reviewer #1

Major points:

Since CAPON deficiency seem to selectively affect hippocampal neurons, functional analysis in terms of behavioral studies may be warranted (if possible). Especially, since this is the first report of CAPON's role in Alzheimer's disease and related tauopathies, if the authors have done or likelihood of performing any behavioral/cognitive analysis, it may be important to include to strengthen their findings.

Thank you for your important suggestion.

Because expression level of endogenous CAPON is higher in hippocampus than other region, it can be assumed that CAPON has more important role in hippocampus and CAPON deficiency has larger effects on hippocampal neurons. However, in this report, since we focus on pathological effects of CAPON-overexpression or deficiency, behavioral analysis is beyond the scope of this study. We promise that we will test cognition under such conditions in the forthcoming publication in the future.

Minor points:

1. Figure 3 can be moved to the supplementary figures since it is mainly showing the validation of AAV-overexpression model.

We have moved Figure 3 (of previous version) to supplemental figures.

2. There are some imperfections in the way the blots are presented. For example, black lines surrounding the blots are missing sometimes, which makes it inconsistent.

We have made figures of blots complete.

3. Specify which co-localizations are shown by Duolink in the figures, so that it will be easier for the easy readability.

We have specified the proteins detected by Duolink in the figures.